# On the importance of the humidity flux for the surface mass balance in the accumulation zone of the Greenland Ice Sheet

Laura J. Dietrich[1], Hans Christian Steen-Larsen[1], Sonja Wahl[1,3], Anne-Katrine Faber[1], and Xavier Fettweis[2]

[1]Geophysical Institute, University of Bergen, and Bjerknes Center for Climate Research, Norway
[2]University of Liège, Belgium
[3]School of Architecture, Civil and Environmental Engineering, Ecole Polytechnique Fédérale de Lausanne, Switzerland

**Correspondence:**
Laura J. Dietrich (Laura.dietrich@uib.no)
Hans Christian Steen-Larsen (Hans.Christian.Steen-Larsen@uib.no)

**Abstract.** It is highly uncertain how the humidity flux between the snow surface and the atmosphere contributes to the surface mass balance (SMB) of the interior Greenland Ice Sheet (GrIS). Due to sparse observations, evaluations of the simulated humidity flux are limited. Model-based estimates of the humidity flux contribution to the SMB are, therefore, unconstrained and even disagree in magnitude and sign. In this study, we evaluate the regional climate model MAR at the EGRIP (East Greenland Ice-Core Project) site in the accumulation zone of the GrIS. We use a combined dataset of continuous one-level bulk estimates of the humidity flux covering the period 05/2016–08/2019 and eddy-resolving eddy-covariance humidity flux measurements from all four summer seasons. In summer, we document a bias of too little sublimation (-1.3 W m$^{-2}$, -1.65 mm w.eq.) caused by a cold bias in both air and surface temperature leading to a reduced humidity gradient. In winter, MAR overestimates vapor deposition by about one order of magnitude. This is a consequence of an overestimated temperature gradient in too stable atmospheric conditions compared to observations. Both systematic errors cause a large discrepancy in the annual net humidity flux between the model and observations of -9 mm w. eq. yr$^{-1}$. Remarkably, the simulated net annual humidity flux contributes positively to the SMB, contrary to observations documenting a net sublimation flux. We correct the systematic errors by applying a simple but effective correction function to the simulated latent heat flux. Using this correction, we find that 5.1 % of the annual mass gain at the EGRIP site sublimates again, and 4.3 % of the total mass gain is deposited vapor from the near-surface air. The estimated net humidity flux contribution to the annual SMB is about -1 % (net sublimation) compared to +5.6 % for the uncorrected simulation. In summer, the corrected MAR simulation shows that vapor deposition accounts for 9.6 % of the total mass gain and that 31 % of the total mass gain at the EGRIP site sublimates again. The net fluxes contribute to -32 % of the summer SMB. These results demonstrate that the humidity flux is a major driver of the summer SMB in the accumulation zone of the GrIS and highlight that even small changes could increase its importance for the annual SMB in a warming climate.

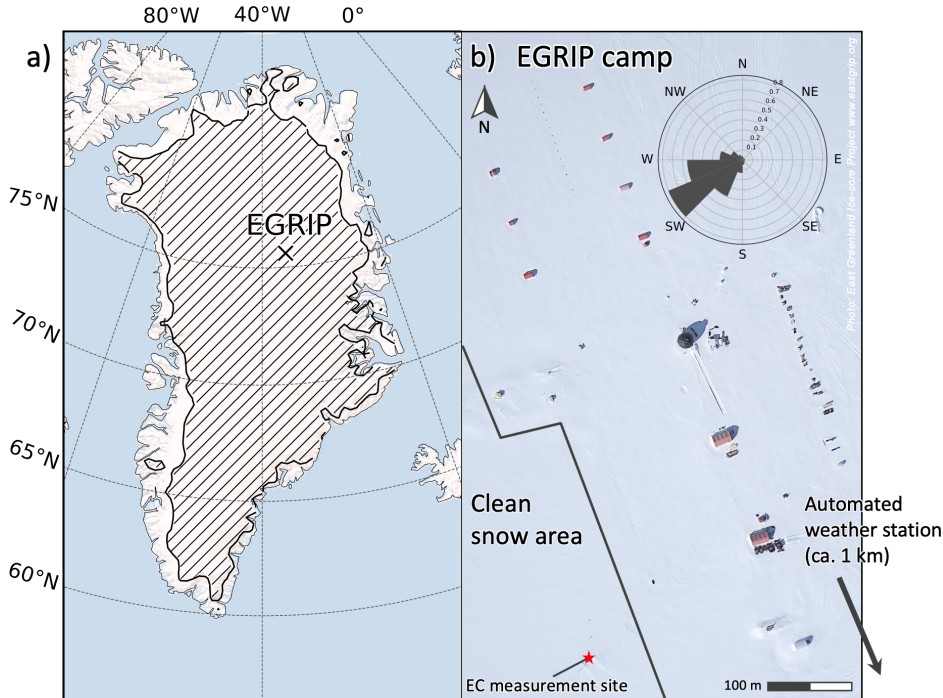

**Figure 1.** Model domain and accumulation zone ($SMB_{2016-2019} > 0$, hatched area) for the MAR model simulation (a) and areal overview of the EGRIP (East Greenland Ice-Core Project) field site (b). The wind rose in (b) shows the normalized distribution of wind directions in the observational period (5/2016–8/2019).

## 1    Introduction

The Greenland Ice Sheet (GrIS) is the second-largest freshwater storage on Earth and loses mass at an increasing rate contributing to about 13.5 mm global mean sea level rise since 1992 (Fox-Kemper et al., 2021). The SMB and its drivers play an essential role in this mass loss (Mouginot et al., 2019). Quantifying all processes influencing the SMB is key to predicting the evolution of the GrIS in a warming climate. The humidity flux directly impacts the SMB by removing snow through evaporation/sublimation or adding mass by condensation/vapor deposition. In the accumulation zone of the GrIS (Fig. 1 a), where temperatures typically stay below the freezing point throughout the year, the only process that transports mass from the ice sheet to the atmosphere is sublimation of surface snow and snow particles in the air. In addition, long surface exposure times in the accumulation zone raise the potential of the humidity flux to impact the snow structure (Casado et al., 2021) and its isotopic composition (Wahl et al., 2022). Accurately simulating the SMB of the accumulation zone of the GrIS and the surface snow properties thus requires an accurate representation of the surface humidity flux in climate and snowpack models.

Regional climate model studies suggest that the annual contribution of the humidity flux to the SMB of the GrIS is minor. This is because the low temperatures above the GrIS lead to a relatively small humidity flux and because of the partial

compensation of sublimation and vapor deposition (Cullen et al., 2014). However, in a warming climate, the counteracting contributions of sublimation and vapor deposition may shift (Boisvert et al. (2017), Zolles and Born (2021)) and as the increasing temperatures amplify the humidity flux, they might gain importance in the future SMB.

Estimating the current contribution of the humidity flux to the SMB requires reliable observational datasets that span at least
a few years. However, observing the turbulent humidity flux in the accumulation zone of the GrIS is challenging due to its remote location and cold and dry atmospheric conditions. Reliable datasets are therefore sparse with low temporal and spatial coverage, and only a few datasets of humidity flux records span multi-annual periods. The humidity flux can be estimated from measuring the humidity gradient and other meteorological parameters following a bulk approach or by using the eddy-resolving eddy-covariance method (EC, first described in Baldocchi (1988)). Generally, the humidity flux is reported as the
latent heat flux (LHF) which is the energy flux during a phase change of water. Note that both used terms LHF and humidity flux in this study are interchangeable. Box and Steffen (2001) used observed meteorological variables from 20 GC-NET automated weather stations (AWS, Steffen et al. (1996)) in the accumulation zone of the GrIS during 1995-1999. Based on the total ice sheet accumulation estimate from Ohmura et al. (1999), they found sublimation to be responsible for either 12 % or 23 % of precipitation loss for following a two-level gradient bulk method, and a one-level bulk method, respectively. Cullen et al.
(2014) used bulk estimates of the LHF from two years (2000-2002) of meteorological observations in the accumulation zone of the GrIS and estimated the mass gain from the humidity flux to be less than 2 % of the annual accumulation. The wide range of these estimates demonstrates that it is challenging to reduce uncertainties in the humidity flux contribution to the SMB of the GrIS accumulation zone due to the sparse spatial coverage of observations.

Most of the available multi-annual records of the LHF are indirectly derived from meteorological observations by calculating the LHF following a one- or two-level bulk estimation method. Bulk estimates are based on the Monin-Obukhov similarity theory, which has limited accuracy under stable atmospheric conditions (Schlögl et al. (2017); Cullen et al. (2007)). Previous observations from the accumulation zone found the atmospheric conditions to be primarily stable in winter and weakly unstable during summer (Cullen and Steffen, 2001). Therefore, using a bulk Monin-Obukhov similarity approach to estimate
the LHF introduces large uncertainty in the contribution of the humidity flux to the SMB for locations such as the GrIS or Antarctic Ice Sheet. Indeed, Town and Walden (2009) found for measurements at the South Pole station that the bulk approach underestimates the LHF by 40–60 % compared to observations obtained using the eddy-resolving EC method. Sigmund et al. (2022) observed a three times smaller flux using a bulk approach compared to the EC method during a storm period at the Syowa S17 station in East Antarctica. They find a flux underestimation using a bulk approach of 16–20 % with blowing snow
turned off in their simulations and even 70–87 % when blowing snow is enabled. Blowing snow is initially deposited snow that is blown up by wind and that is in suspension in the air up to several tens of meters above the ground. In addition, the bulk method requires knowledge of the aerodynamic roughness length, which is usually unknown, and assumptions about the wind profile must be made. The roughness length is a conceptual parameter that describes the estimated influence of turbulence on the vertical transport of moisture, momentum, or heat, assuming a logarithmic wind profile. In the EC method, a large part of

the turbulent eddy spectrum is resolved directly, and no assumptions regarding the wind profile are required to obtain the LHF. By observing fluctuations in the vertical wind $w$ and the specific humidity $q$, the LHF can be derived by $LHF = \text{cov}(w, q) \cdot L_s$, where $L_s = 2.831 \cdot 10^6 \, \text{J} \, \text{kg}^{-1}$ is the latent heat of sublimation. The underlying assumptions of the EC method are steady-state conditions during the measurement integration interval, in particular, a stationary, horizontal homogeneity of flow, and well-developed turbulence (Foken, 2021). The eddy-resolving EC method is a good alternative for observing the humidity flux in the accumulation zone of the GrIS, with commonly smaller errors under the prevailing neutral to slightly stable conditions compared to the bulk method, and has been successfully used to estimate turbulent fluxes in Greenland in previous studies (Van Tiggelen et al. (2020), Miller et al. (2017)).

To compensate for the sparse observations and to obtain spatial coverage in the accumulation zone, climate model simulations of the SMB are indispensable. Thus, most of the current estimates of the humidity flux contribution rely solely on model simulations. However, these model simulations are unconstrained as the parameterizations of humidity exchange processes at the snow surface for the accumulation zone have not been evaluated on neither intra- nor inter-annual time scales. Thus, the role that the humidity flux plays for the SMB of the interior GrIS remains unconstrained, if not completely unknown.

This study addresses the uncertainty in regional climate model estimates of the humidity flux contribution to the SMB in the accumulation zone of the GrIS. We use a novel EC dataset of the humidity flux from the EGRIP drilling site during four summers (June and July 2016–2019) to evaluate the state-of-the-art regional climate model MAR (described in Section 2) for polar regions. Based on the evaluation, we apply a simple correction function, which allows us to constrain the model simulations and to provide an improved estimate of the seasonal and annual humidity flux contribution to the SMB at the EGRIP site. Note that in this study, an upwards directed humidity flux (sublimation) is defined positive, causing a negative contribution (mass loss) to the SMB.

## 2   Data and Methods

**Regional climate model MAR**

To investigate the hydrological cycle over the GrIS, we use version v3.11 of the hydrostatic regional climate model MAR (Modèle Atmosphérique Régional, e.g., Fettweis et al. (2017)) and simulate the surface processes, such as snow melt, sublimation, and vapor deposition, refreezing, changes in snow optical properties and snow texture, as well as mass and energy balance in the period 2016–2019. MAR is a well-established model for SMB simulations over the GrIS (e.g. Fettweis et al. (2020), Goelzer et al. (2020)). We run MAR on a vertical resolution of 30 atmospheric layers between 1 m and the top of the atmosphere (0.1 hPa) with an increased resolution towards lower altitudes. We use a horizontal resolution of 30 km on a 66 x 113 grid points domain (Fig. 1 a), which is a sufficient resolution to investigate the flat and orographically smooth top of the GrIS. For the comparison with the observational data from the EGRIP site the nearest grid cell is chosen. The simulations are calculated for a time step of 180 seconds, and MAR's output is given in hourly averages. The simulation is initialized and

forced at its boundaries with 6-hourly data of the ERA-5 reanalysis product (Hersbach et al., 2020). The atmospheric model is coupled to the 1-D Surface Vegetation Atmosphere Transfer scheme SISVAT (Soil Ice Snow Vegetation Atmosphere Transfer, Fettweis et al. (2005)), which simulates the snow-atmosphere interactions of energy and mass. The snowpack and snow properties are simulated based on an early version of the CROCUS snow model implemented in SISVAT. The blowing-snow module (Gallée et al. (2001), Amory et al. (2021)) was turned off in this study to exclude local redistribution of deposited snow from the SMB simulation at the EGRIP site.

**Latent heat flux parameterization in MAR**

The LHF in MAR is calculated using a one-level bulk parameterization. At the snow surface, saturation with respect to the snow surface temperature and a wind speed of zero is assumed. The LHF is calculated by

$$LHF = -\rho L_s \kappa^2 \frac{u}{\ln\left(\frac{z_u}{z_{u,0}}\right) - \Psi_u} \cdot \frac{q - q_s}{\ln\left(\frac{z_q}{z_{q,0}}\right) - \Psi_q} \tag{1}$$

where $\rho$ is the air density, and $\kappa \approx 0.4$ the von Kármán constant. $u$ is the wind speed, and $q$ the air specific humidity at the heights $z_u = 2\,\text{m}$ and $z_q = 2\,\text{m}$, respectively. In the MAR simulations, for lack of a better parameterization, we use a fixed roughness length for momentum of $1.3 \cdot 10^{-4}$ m throughout the entire simulation period. This value corresponds to the median of the roughness length estimated from EC measurements at EGRIP in the summer of 2019 (Steen-Larsen and Wahl, 2021) and agrees with EC measurements from the katabatic wind zone of the Antarctic Ice Sheet ($z_0 = 1.6 \cdot 10^{-4}$ m, Van den Broeke et al., 2005). The roughness length of moisture $z_{0,q}$ is derived following the parameterization by Andreas (1987) for snow surfaces. The stability correction functions for momentum $\Psi_u$ and for moisture $\Psi_q$ are calculated following Holtslag and De Bruin (1988) for stable atmospheric conditions and assuming $\Psi_q = \Psi_u$. For unstable conditions, the Businger-Dyer representation is used, described in Paulson (1970):

$$\Psi_u = 2 \cdot \ln\left(\frac{1+x}{2}\right) + \ln\left(\frac{1+x^2}{2}\right) - 2 \cdot \tan^{-1}(x) + \frac{\pi}{2}$$

and

$$\Psi_q = 2 \cdot \ln\left(\frac{1+x^2}{2}\right)$$

where $x = (1 - \gamma/\zeta)^{\frac{1}{4}}$ with the dimensionless stability parameter $\zeta = z_u/L$, where $L$ is the Monin-Obukhov length and $\gamma = 16$ an empirically derived constant.

**Meteorological data**

Meteorological observations are obtained from an AWS as part of the Programme for Monitoring of the Greenland Ice Sheet (PROMICE, Fausto et al. (2021), Tab. A1). The AWS was installed in May 2016 and is regularly maintained. It provides continuous observations of wind speed, humidity, surface temperature, 2 m temperature, and solar and thermal radiation fluxes. In 2016–2019 the sensors had an average height of 2.3 m [2.6 m to 1.8 m] above the snow surface, and the meteorological data is compared to the model output at 2 m without correcting for the height difference, assuming it is insignificant.

**Atmospheric eddy-covariance system**

We evaluate the surface humidity flux in MAR using a dataset of eddy-resolving EC measurements from the EGRIP site (Steen-
135  Larsen et al., 2022). The EC system consists of a CSAT3 wind sensor combined with a KH20 hygrometer (both by Campbell
Scientific) which are installed at a height of 1.80 m facing the prevailing wind direction (240°, Fig. 1 b). The EC system
measures the three-dimensional wind speed ($u,v,w$), and the water vapor density $q$ (kg m$^{-3}$) at a high sampling frequency of
20 Hz. In that way, the EC method resolves the turbulent eddies and measures the turbulent transport of moisture.

The integration time of the measurement should be chosen in the so-called spectral gap between frequencies associated with
turbulence and those associated with the mean flow (Stull, 1988). On the one hand, the integration time should be long enough
to capture most of the turbulent eddy frequency spectrum. On the other hand, the integration time has to be short enough to
fulfill the assumption of steady-state conditions and, thus, to exclude variations in the mean flow. The integration time in the
datasets used in this study was 10 minutes for the measurements in 2016–2018 and 30 minutes in 2019. Based on the raw EC
measurements in 2018, an integration time of 30 minutes was found to be the optimal choice at EGRIP, covering the majority
of turbulent frequencies. However, differences in 10- and 30-minute integration time were minor using the 2018 raw eddy-
covariance dataset, indicating that both integration times lie well within the spectral gap. The available 10-minute EC dataset
from 2016–2018 was averaged to half-hour frequencies prior to publication to fit the frequencies of the 2019 dataset with 30
minutes integration time. The EC data were filtered for outliers (LHF$_E C$<-20 or >40 W m$^{-2}$). We average the EC data from
2016–2019 to hourly values for compatibility with the model data, resulting in a total amount of 5304 data points.

**2.1 Site description**

All measurements for the model evaluation were carried out as part of the deep ice core drilling project EGRIP (East Greenland
Ice-Core Project). The EGRIP drilling site is located at 75° 38' N and 36° 00' W in the interior of the GrIS accumulation zone
at an approximate height of 2700 m above sea level. The local time (LT) is two hours behind the Coordinated Universal Time
(UTC). Meteorological observations are provided by an AWS located about 1 km southeast of the EGRIP camp (Fig. 1 b). The
EC system is set up in a dedicated clean snow area upstream in the prevailing wind direction from the EGRIP camp.

During the summer months of 2016–2019, the prevailing wind direction was west to north-west with an average wind
speed of 4.6 m s$^{-1}$ (daily average between 1.1 and 8.9 m s$-1$), and the average 2 m air temperature was -9.4 °C (daily average
between -22.6 and -1.9 °C) with the average diurnal cycle in summer (June and July) spanning -14.7 – -8 °C. All four years
have similar meteorological conditions (Fig. A1), but 2016 and 2019 were slightly warmer and moister during summer, leading
to generally higher LHF.

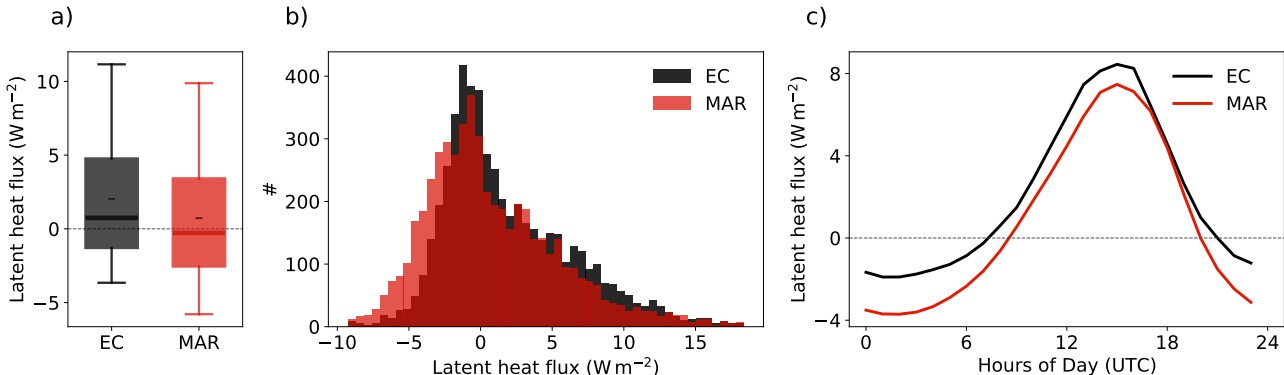

**Figure 2.** Distribution (a,b) and diurnal cycle (c) of the observed (EC, black) and simulated (MAR, red) hourly latent heat flux during all summers in the observational period (June and July 2016–2019) at the EGRIP location. In (a), black dashes (-) denote the mean, thick lines denote the median, boxes denote the 25th–75th percentile, and whiskers denote the 5th–95th percentile. Hours of the day in (c) are given in UTC, and the LT corresponds to UTC-2 h.

## 3 Results

### 3.1 Evaluation of the regional climate model MAR

We evaluate the simulated LHF at EGRIP against the EC observations for all summers (June and July) in 2016–2019 in Fig. 2.
MAR systematically underestimates the LHF with a mean bias of -1.3 W m$^{-2}$. In all four summers, the bias is consistently negative and independent of the time of day. Besides the bias, MAR captures the diurnal cycle well (Fig. 2 c). The simulated LHF has a similar diurnal range, spanning from -3.7 to 7.5 W m$^{-2}$, and the diurnal maximum and minimum are aligned with the observations. During summer, the simulated daily mean LHF has a standard deviation of 1.79 W m$^{-2}$, similar to the observations ($STD_{Obs, daily\ mean} = 1.92$ W m$^{-2}$). The daily mean values of the simulated and observed LHF are only weakly
correlated ($R_{MAR-Obs} = 0.27$). This goes along with a high non-systematic error in the daily mean LHF simulations with a root-mean-square error (RSME) of 2.66 W m$^{-2}$, equivalent to 1.39 times the observed standard deviation. MAR is only forced with the reanalysis at the domain's boundaries every six hours, and therefore, the exact timing of weather events may be shifted. On sub-diurnal time scales, MAR performs slightly better during the night hours (6 pm - 6 am LT) with a correlation of $R_{MAR-Obs} = 0.25$ compared to $R_{MAR-Obs} = 0.20$ during the day hours (6 am - 6 pm LT, Fig. A2). Despite the high non-
systematic error, MAR captures the distribution of the LHF remarkably well in the four summers of the observational period (Fig. 2 a,b). Similar to the observations, the distribution of the simulated LHF has a slightly right-skewed shape, spanning a range of -10 to 20 W m$^{-2}$, with 50 % of the LHF being between -2.5 and 3.4 W m$^{-2}$.

Fig. 3 shows the variables that have a direct impact on the LHF (Eq. 1) for the exemplary summer of 2019. The distribu-
tion of the LHF in the summer of 2019 (right panel of Fig. 3 a) is similar to the distribution of the LHF in all four summers

in the entire period 2016–2019, which is shown in Fig. A3. MAR captures the daily wind speed ($R_{MAR-Obs} = 0.76$) and air density ($R_{MAR-Obs} = 0.79$) in both distribution and mean but fails to reproduce the daily specific humidity gradient $\Delta q$ ($R_{MAR-Obs} = 0.28$), which is defined as the difference between $q$ at 2 m height and the saturation specific humidity at the surface $q_{s,sat}$. While in the observations, $\Delta q$ is mostly negative (i.e. $q_{s,sat} > q_{2m}$), MAR simulates a net zero $\Delta q$ leading to a bias in the humidity flux. Similar to the LHF, although the daily values of $\Delta q$ differ from the observations, its diurnal cycle and the shape of the distribution are captured.

Estimating the humidity flux contribution to the SMB requires LHF simulations in all seasons. However, there are no year-round eddy-resolving EC flux measurements of the LHF available for the EGRIP site. To evaluate the LHF simulation in MAR beyond the summer months, we estimate the LHF from meteorological variables measured by the PROMICE AWS using a similar one-level bulk parameterization method as implemented in MAR. For the calculations we use a constant roughness length of $1 \cdot 10^{-5}$ m. This is a substantially smaller value than the median of the derived roughness length from the EC measurements in summer 2019 ($1.3 \cdot 10^{-4}$ m) but it provides the best agreement between the bulk estimates and the direct EC observations of the LHF in all summers 2016-2019. The LHF of the EC observations and the bulk estimate has a correlation of $R_{MAR-Obs} = 0.72$ after removing the diurnal cycle and an RMSE of 2.3 W m$^{-2}$ (Fig. A4). This gives confidence in the bulk estimates despite the weakly stable conditions. Like in summer, the simulated LHF in MAR is biased towards vapor deposition throughout the entire season (Fig. 4 a, Fig. 5 c). Nevertheless, MAR simulates a shape of the seasonal cycle similar to the observations with a monthly mean value correlation of 0.88 (Fig. 4 a). After removing the seasonal cycle, the correlation of the monthly mean values reduces to 0.31, with MAR performing better in spring ($R_{MAR-Obs,MAM} = 0.59$) and autumn ($R_{MAR-Obs,SON} = 0.80$) than in summer ($R_{MAR-Obs,JJA} = -0.19$) and winter ($R_{MAR-Obs,DJF} = 0.33$). Note that these seasonal correlations are only based on four years, resulting in a total of 12 different monthly values out of each season. Daily mean values (after removing the seasonal cycle) have a correlation with the bulk estimate of $R_{MAR-Obs} = 0.42$ for the entire observational period.

The monthly mean values of MAR are generally lower than the LHF estimated with both methods bulk and EC (Fig. 5). Apart from an offset, the shape of the simulated LHF annual distribution differs from the bulk estimate (Fig. 5). The simulations have a more right-skewed shape than the bulk estimates, as the occurrence of negative fluxes is overestimated, and the occurrence of positive and small fluxes, close to zero, is underestimated. Contrary to summer, in winter, the magnitude of the simulated LHF is about one order of magnitude larger than the bulk estimate on both hourly (Fig. 6) and daily time scales (Fig. 4 b). The systematic error in winter is a factor rather than an offset.

We analyze the hourly and daily variability in MAR and observations in Fig. 6 for the exemplary winter of 2019. The distribution of the LHF in winter 2019 (right panel of Fig. 6 a) is similar to the total distribution of the LHF in all winter months in the entire period 2016-2019, which is shown in Fig. A5. The bulk estimate of the LHF has a low daily variability with mostly negative values close to zero for most of the winter except for isolated events. On the contrary, MAR simulates a

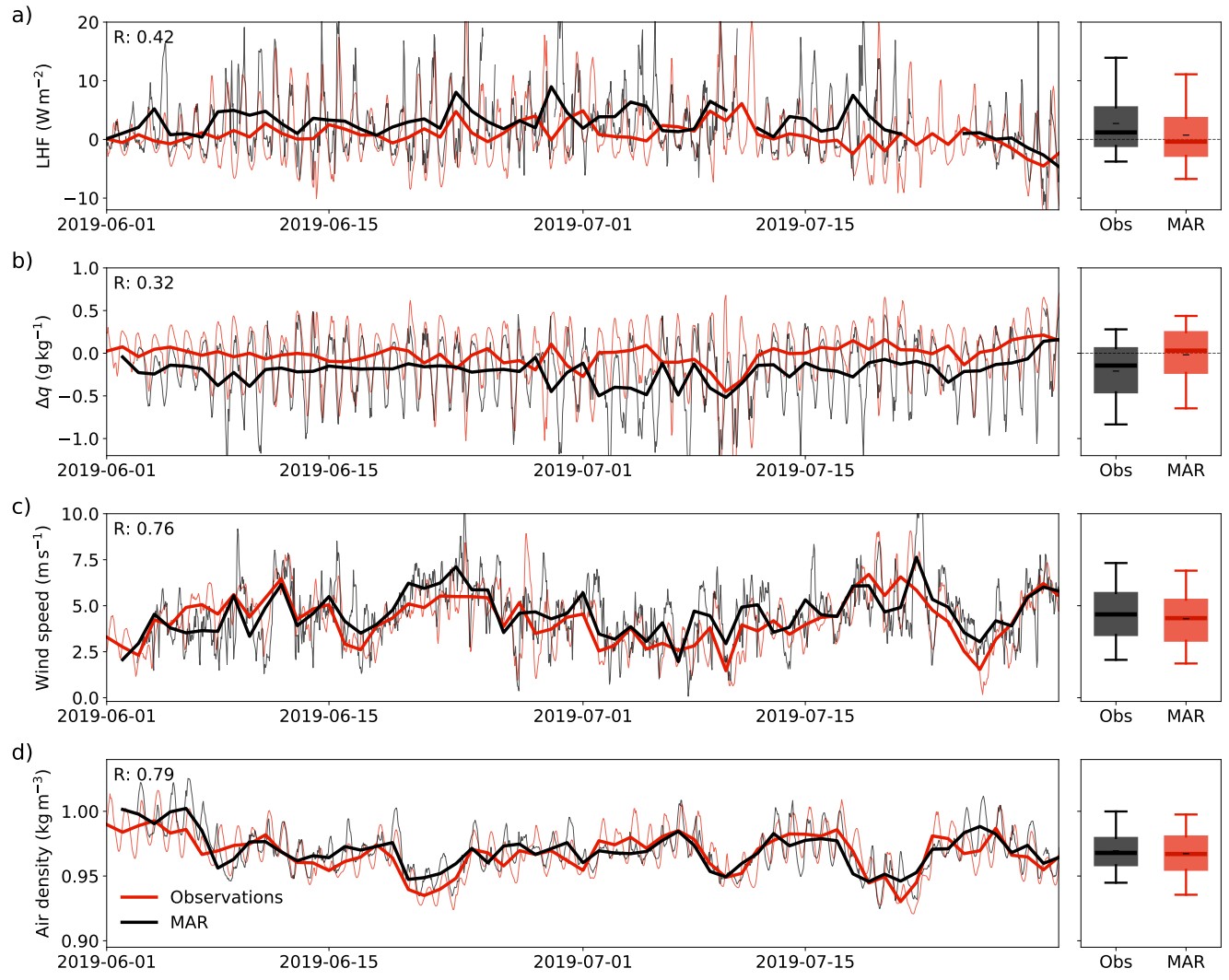

**Figure 3.** Time series of the latent heat flux (LHF, a), specific humidity gradient ($\Delta q$, b), wind speed (c), and air density (d) in the period 06/2019–07/2019 from observations (Obs, black) and MAR (red). The LHF is based on EC measurements, while the specific humidity gradient, wind speed, and air density are obtained from the AWS. The bold lines show daily averages, hourly data is plotted in thin lines. The boxplots (right) show the distribution of the hourly data presented in the time series plots (left). In the boxplots, black dashes (-) denote the mean, thick lines denote the median, boxes denote the 25th–75th percentile, and whiskers denote the 5th–95th percentile.

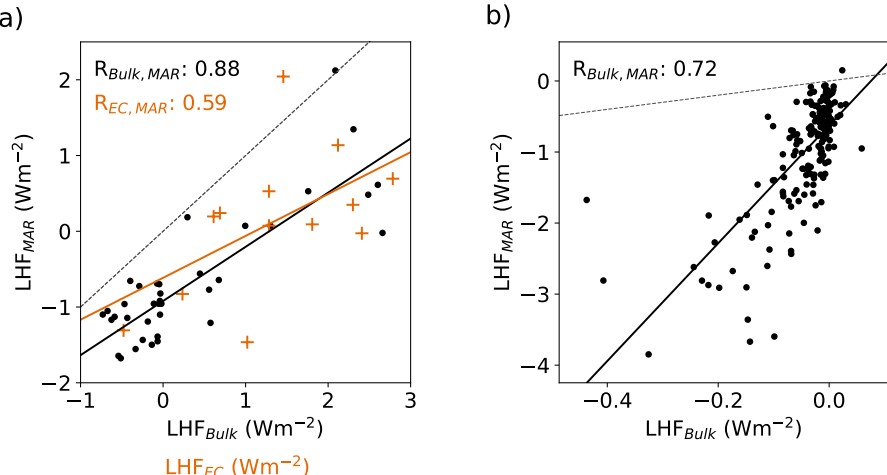

**Figure 4.** a) Comparison of the LHF monthly mean values for the bulk calculation (black dots) and EC measurements (orange crosses) to the MAR simulation. Note that there are only EC data available for May, June, July, and August. b) Comparison of the LHF daily mean values for the bulk calculation for all winters (December, January) in 2016-2019. Note that in winter the daily net fluxes are exclusively depositional. The grey dashed lines show the one-to-one lines.

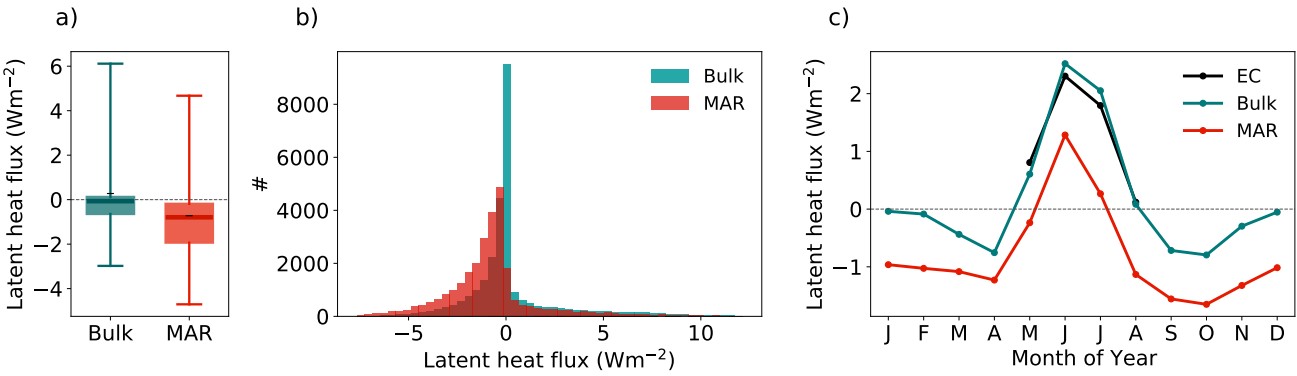

**Figure 5.** Distribution (a,b) and seasonal cycle (c) of the hourly latent heat flux calculated from observations (bulk, turquoise) and simulated (MAR, red) during the entire observational period (05/2016–08/2019). In (a), black dashes (-) denote the mean, thick lines denote the median, boxes denote the 25th–75th percentile, and whiskers denote the 5th–95th percentile.

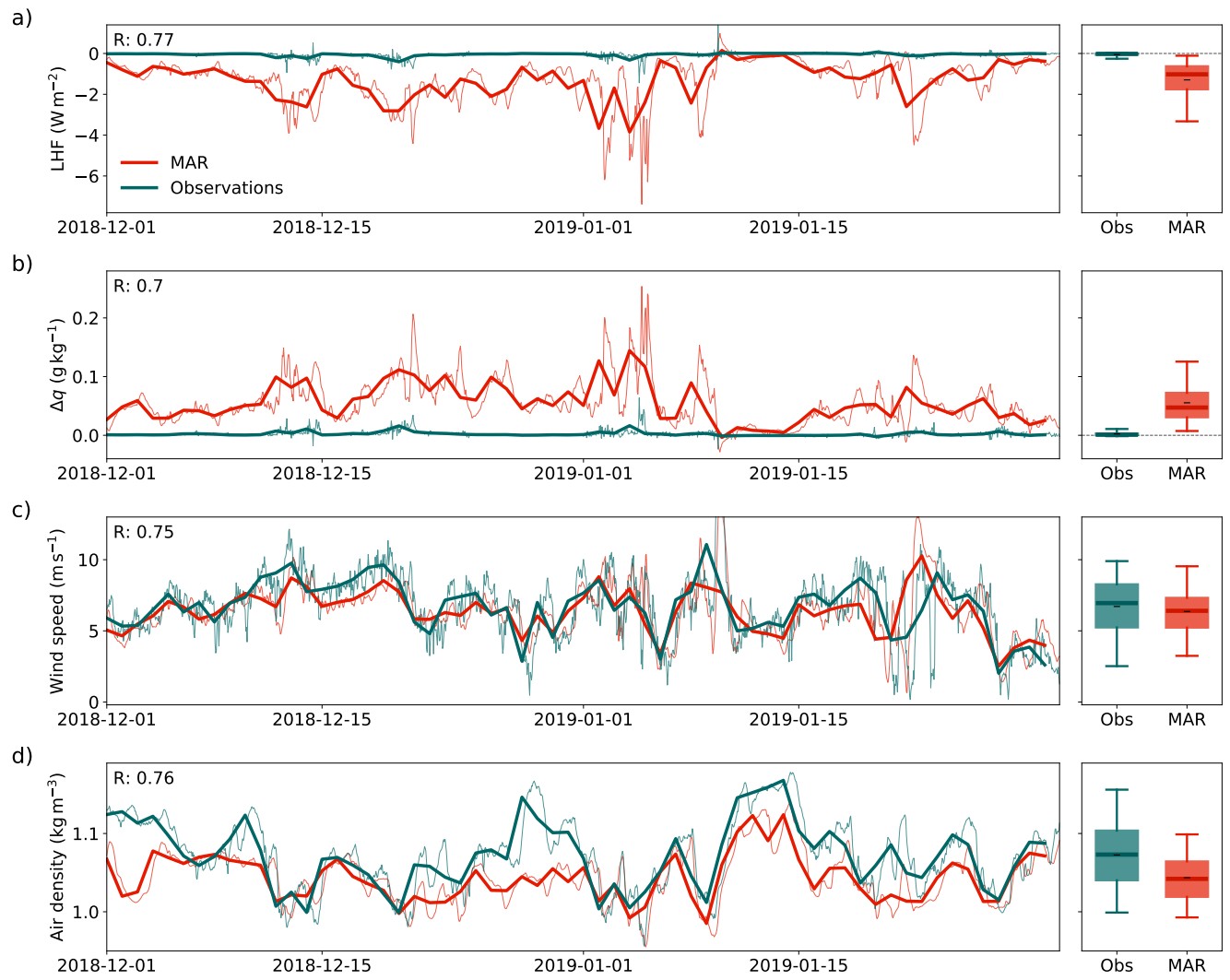

**Figure 6.** Time series (left) and distribution (right) of the latent heat flux (LHF, a), specific humidity gradient ($\Delta q$, b), wind speed (c), and air density (d) in the period 12/2018–01/2019 from observations (Obs, turquoise) and MAR (red). The bold lines show the daily averages of the hourly data (thin lines). The boxplots (right) show the distribution of the hourly data presented in the time series plots (left). In the boxplots, black dashes (-) denote the mean, thick lines denote the median, boxes denote the 25th–75th percentile, and whiskers denote the 5th–95th percentile.

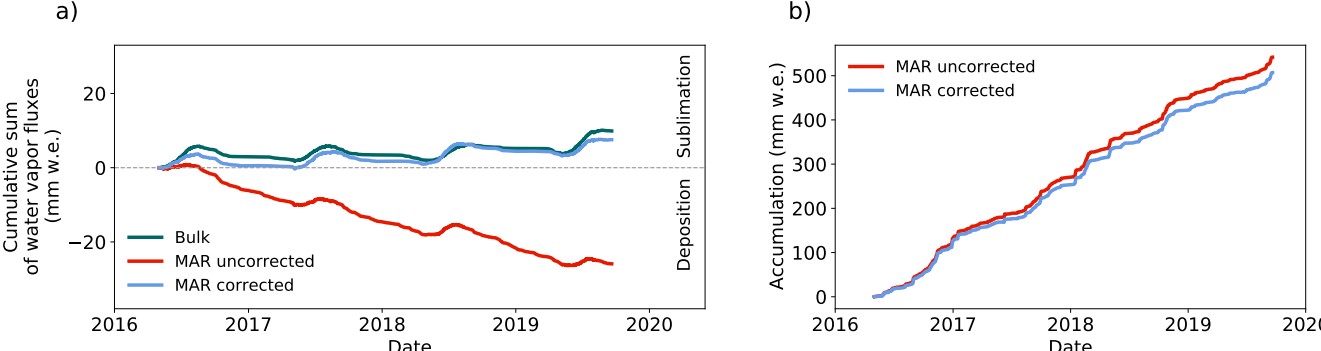

**Figure 7.** Left: Cumulative sum of the humidity flux from bulk estimates using meteorological observations from the PROMICE AWS (dark green, bulk), simulated by MAR (red) and the corrected MAR simulation (blue) for the time period. Positive values correspond to a surface mass loss due to sublimation. Right: Cumulative sum of the simulated total accumulation (snowfall + vapor deposition - sublimation) in MAR.

relatively strong depositional flux. Despite the strong overestimation, MAR is consistent in the timing of the observed events, such that the daily mean correlation of the LHF is high ($R_{MAR-Obs} = 0.77$). Like in summer, the specific humidity gradient can explain the major part of the LHF difference in winter between the observations and MAR (Fig. 6 b). Additionally, $\rho$ shows a small bias (Fig. 6 d). The bias in $\rho$ is caused by equal contributions of both a bias in pressure and temperature.

## 3.2  Improved estimate of the SMB contribution

The evaluation in Section 3.1 shows that, in summer, MAR has a negative offset bias in the LHF while simulating a similar variability as observed. In winter, this offset diminishes, but MAR overestimates the magnitude of the LHF and, thus, overestimates both the variability and the total vapor deposition. These seemingly small systematic errors in the humidity flux lead to a mean difference of -9 mm w. eq. yr$^{-1}$ in its contribution to the SMB at the EGRIP site (Fig. 7). In fact, MAR simulates a positive contribution (net vapor deposition) to the annual SMB, while the observations show a slightly negative contribution (net sublimation). To obtain an improved estimate of the humidity flux contribution to the annual SMB, we propose a simple linear correction function $f(LHF_{\text{MAR, corr}}) = m(\overline{q_{s,sat,\text{monthly}}^{-1}}) \cdot LHF_{\text{MAR}} + b(\overline{q_{s,sat,\text{monthly}}})$ based on the monthly averages of $q_{s,sat}$ to correct for systematic errors in the simulated LHF (Listing 1).

The parameters $m$ and $b$ (Fig. 8) account for the two different types of systematic errors in MAR described above. The systematic error in the flux magnitude is corrected by $m = \overline{q_{s,sat,\text{summer}}^{-1}} \cdot \left( \overline{q_{s,sat,\text{monthly}}^{-1}} - \overline{q_{s,sat,\text{summer}}^{-1}} \right)$ that depends on the inverse of $q_{s,sat}$ and is normalized so that it varies between 1 in summer (June and July) and 0.1 in winter (December and January). The flux offset is corrected by $b = \text{BIAS} \cdot \overline{q_{s,sat,\text{summer}}^{-1}} \cdot \overline{q_{s,sat,\text{monthly}}}$ that depends directly on $q_{s,sat}$, and varies between the observed offset of $\text{BIAS} = 1.3$ in summer (June and July) and 0.1 on average in winter (December and January), set to zero on January 1st. The overbars indicate the mean.

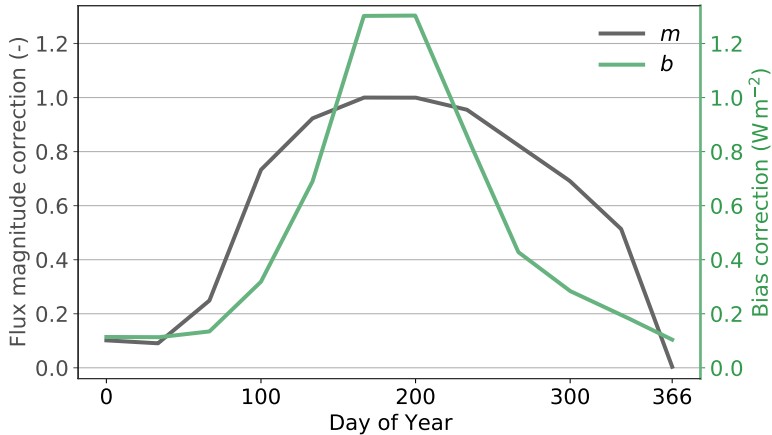

**Figure 8.** Annual cycle of the parameters $m$ and $b$ of the linear correction function for the simulated LHF.

The systematic error in the LHF simulation is mainly caused by systematic errors in the surface and 2 m temperature and their difference that affects the near-surface humidity gradient (see Sec. 3.2.1). Therefore, the correction function of the LHF simulation is based on $q_{s,\mathrm{sat}}$, which has a nonlinear dependence on the surface temperature (Clausius-Clapeyron). The factor $m$ depends on the inverse of $q_{s,\mathrm{sat}}$ and is normalized so that it varies between 1 in summer (June and July) and 0.1 in winter (December and January). Thus, there is no effect on the flux magnitude in summer but a flux magnitude reduction in winter by one order of magnitude. The offset correction $b$ depends directly on $q_{s,\mathrm{sat}}$, and $b$ varies between the observed bias value of 1.3 in summer and 0.1 (zero bias on January 1st) in winter.

As a result of the factor correction, the vapor deposition during winter is strongly reduced, leading to a smaller mass gain (Fig. A8 c). During summer, the bias correction function shifts the LHF towards enhanced sublimation, and thus, the mass loss is increased. We use the corrected LHF simulations in MAR and calculate the humidity flux contribution to the SMB at the EGRIP drilling site in the period 2016–2019 as well as the range for the individual years to estimate the inter-annual variability. By applying the correction, we find that contrary to MAR's uncorrected LHF simulation and previous estimates (e.g., Fettweis (2007)), the net humidity flux at EGRIP is not negative but positive (Fig. 7 a), i.e., causes a net mass loss equivalent to -1 % of the annual SMB, while the uncorrected simulation shows a mass gain equivalent to +5.6 % of the annual SMB. In the corrected simulation, 5.1 % [4 % to 6 %] of the annual mass gain (snowfall + vapor deposition) sublimates again, and 4.3 % [3.2 % to 5.3 %] of the mass gain is deposited water vapor from the air. During summer, the portion of vapor fluxes in the SMB is much larger than in the annual mean with 31 % [26 % to 34 %] of the summer mass gain (snowfall + deposition) sublimating again. Vapor deposition accounts for 9.6 % [7.4 % to 12 %] of the mass gain during summer in the corrected simulation (Fig. A6). The simulated net humidity flux contributes to 32 % [23 % to 37 %] of the summer SMB. Note that these numbers are not based on

simulations from a corrected MAR model version, but only post-corrections were applied. A potential feedback of the change in LHF has no impact on the simulated precipitation amount. Our results demonstrate that the humidity flux, in particular sublimation, has a major contribution to the summer SMB (Fig. A6), and that an accurate humidity flux representation in models is important for the simulation of the SMB over multi-annual time scales (Fig. 7 b).

### 3.2.1 Discussion

Regional climate models are key to estimating the SMB of the GrIS, as observations are challenging in polar environments. Our evaluation shows that MAR captures the distribution of the LHF (Fig. A8) as well as the distribution of the wind speed (Fig. A3, Fig. A5) remarkably well when using an appropriate surface roughness length for the interior GrIS. Capturing the distribution of the LHF is for many climate model investigations more important than having a very low hour-to-hour random error. However, MAR has systematic errors in the temperature and near-surface stratification with consequent impacts on the simulated LHF. Even small systematic errors in the humidity flux simulation can have large impacts on the SMB contribution on seasonal and annual time scales, as well as potential impacts on the simulated snow surface properties. This study addresses the temperature-driven systematic errors by post-correcting the LHF based on the surface saturation specific humidity to account for the non-linear impact of temperature biases on the humidity flux. While this approach provides a good estimate of the potential long-term error in the LHF simulation as part of the SMB, the impacts of a changed LHF on the simulation itself are not considered, such as on the radiative surface budget or the surface-near humidity. We conclude that MAR is a well-performing tool for humidity flux simulations on climatological timescales in the accumulation zone of the GrIS, but the cause of systematic errors in the 2 m temperature and the surface-near temperature gradient needs to be understood and addressed in the model. However, our results show that even small systematic errors in the humidity flux can have large impacts on the SMB contribution on seasonal and annual time scales, with potential impacts on the snow surface properties. Correcting the seemingly small summer bias of -1.3 W m$^{-2}$ (-1.65 mm w.eq.) in the LHF simulation leads to a three times smaller mass loss due to the humidity flux over the four summers (not shown). This stresses the importance of both accurate observations of the humidity flux on the Greenland Ice Sheet and an accurate representation in surface mass balance models.

The model evaluation in the summers 2016–2019 is based on EC measurements of the humidity flux at EGRIP. Previous studies have shown that estimating the humidity flux with the EC method can be challenging (e.g. Cullen et al. (2007)). While a too stable boundary layer stratification is often a limitation for EC measurements above snow surfaces, the consistent katabatic wind flow at EGRIP leads to mostly unstable to slightly stable stratification (Wahl et al., 2021) and a rather constant wind direction. Combined with very homogeneous and smooth terrain at EGRIP, the conditions are generally suitable for the underlying assumptions of a stationary and horizontal homogeneous flow. Stringent quality control of an EC dataset in 2019 can be found in Wahl et al. (2021), generally showing high data quality under unstable and neutral conditions at EGRIP. However, a surface energy budget has not been set up, and systematic errors in the humidity flux estimates in summer from the EC system cannot be ruled out. Moreover, the caveats of the bulk methods to estimate turbulent humidity fluxes are well-known, and the lack of other flux measurement systems operating at EGRIP in winter makes an independent validation of the quality of these data and

potential error sources, such as the impact of instrumental frost on the record, impossible, and the bulk estimated humidity flux needs to be treated with care.

Errors in the simulated LHF are potentially caused by an erroneous representation of three meteorological variables (Eq. 1): (1) the wind speed, (2) the air density, and (3) the specific humidity gradient. In summer, the bias towards a smaller LHF is primarily driven by a bias in $\Delta q$ towards smaller gradients and, to a lesser degree, by a small negative bias in both air density and wind speed. The bias in the $\Delta q$ is caused by a cold bias in the surface temperature of -1.5 K, affecting the surface saturation specific humidity. The 2 m air temperature has a similarly strong bias (-1.2 K), however, MAR overestimates the relative humidity so that the distribution of the specific humidity at 2 m agrees with the observations. The cold biases in both the surface and the 2 m air temperature are a direct consequence of a negative bias in the downward long-wave radiation flux linked to the cloud scheme implemented in MAR (Fig. A9). Besides the systematic error, $\Delta q$ explains 64 % of the non-systematic error in the daily LHF averages in summer as well.

In winter, the systematic error in the LHF is not a constant offset, like in summer, but a consistent overestimation of the LHF magnitude. Again, the wind speed is captured well by the model ($R_{MAR-Obs} = 0.72$, $RMSE = 0.73\,\mathrm{m\,s^{-1}}$) with only a small bias of $0.18\,\mathrm{m\,s^{-1}}$. The bias in the air density of $-0.027\,\mathrm{kg\,m^{-3}}$ is a direct consequence of, first, a bias in the air pressure of -6.6 hPa and, second, an overestimation of the 2 m air temperature by 3.67 K. However, the simulated air density is only about 2.7 % lower than the observations and is, hence, considered to have a small impact. Like in summer, it is thus mainly $\Delta q$ that causes the systematic error in the LHF in winter. Moreover, $\Delta q$ explains more than 90 % of the non-systematic error of the daily mean values in winter.

The specific humidity gradient is primarily driven by the temperature gradient due to the Clausius-Clapeyron relationship and, to a lesser degree, by the relative humidity. MAR also systematically underestimates the relative humidity during winter (not shown). However, this would counteract the LHF overestimation and is found to be insignificant compared to the impact of the temperature gradient. The simulated temperature gradient is on average overestimated by 2.93 K in winter, causing a strong overestimation of $\Delta q$. This is a consequence of a large bias in the simulated 2 m temperature (3.67 K), while the simulated surface temperature has a smaller positive bias of 0.73 K. Thus, MAR consistently simulates a stable regime with an average positive temperature gradient of 2.82 K. In fact, the observations show a very small temperature gradient (well-mixed neutral stratification) during winter at EGRIP. Contrary to summer, when both MAR and the observations show a slightly stable regime and their flux magnitudes are similar, MAR overestimates the magnitude of the (mostly depositional) flux in winter due to the strong stratification. A plausible explanation for the observed neutral atmospheric conditions during winter could be the dominating effect of mixing by katabatic winds at EGRIP, which might not be accurately captured by MAR.

We find that the systematic error of the LHF simulation in MAR can be explained to a large degree by a systematic error in $\Delta q$. The humidity gradient is not only impacted by the temperature gradient itself but also by the absolute temperature values.

Because of the non-linear relationship between temperature and specific humidity (Clausius-Clapeyron relationship), even if the temperature gradient and the relative humidity were perfectly captured, a bias in both surface and 2 m air temperature would cause a bias in $\Delta q$. Additionally, a bias in the temperature gradient causes a deviating magnitude of the humidity flux. We, therefore, argue that the bias correction $b$ has to depend directly on the saturation specific humidity to characterize the non-linear dependence of the specific humidity on the temperature. Similarly, the factor correction $m$ has to depend on the inverse of the saturation specific humidity because an overestimation of the humidity gradient needs counteracting by smaller values of $m$.

Two choices to set up the simulations were made that have a direct impact on the LHF simulation. First, we found the default roughness length parameterization in MAR unsuitable for the smooth surface in the accumulation zone of the GrIS, producing consistently too high roughness length values and, consequently, reduced wind speeds. We, therefore, set the roughness length for momentum to a constant value of $1.3 \cdot 10^{-4}$ m. There are no measurements of the roughness length at EGRIP available in winter. However, we are confident that the chosen roughness length is suitable for simulations in the accumulation zone of the GrIS year-round as the resulting wind speeds are consistent with the observations (Fig. A7).

Second, we turned off blowing snow in the simulations to avoid local SMB variations and to exclude redistribution of deposited snow from the SMB input. To our knowledge, the blowing snow module in MAR has not yet been evaluated against observations for the Greenland Ice Sheet. By excluding the impact of blowing snow in the simulation, we avoid compensation from biases in MAR's temperature and potential systematic errors in the impact of blowing snow on sublimation. Thus, the mismatch between the model and the observations can be directly attributed to uncertainties in the observations, known temperature biases in MAR, as well as the neglect of blowing snow in the simulation. However, the importance of blowing snow in the surface mass and energy balance at EGRIP has not been quantified. Many studies show that sublimation on blowing and drifting snow particles is a key contributor to the total surface humidity flux in Antarctica (e.g. Palm et al. (2017), Van Den Broeke et al. (2010)). Le Toumelin et al. (2021) find that blowing snow and associated blowing snow sublimation improves MAR's ability to capture the surface energy and mass balance in the coastal Adelie Land, Antarctica, more accurately. Their simulations indicate that blowing snow decreases the LHF. It should therefore be taken into account when estimating the contribution of the humidity flux to the SMB in regions that are prone to blowing snow and with relatively little accumulation, such as the Antarctic accumulation zone. We further stress that observations of the sublimation on blowing snow for the GrIS accumulation zone are needed to quantify the total contribution of the humidity flux to its SMB accurately. During the field observation periods in 2016–2019, observed blowing snow events at EGRIP were rare. We tested the impact on the LHF of turning on the blowing snow module in MAR for the period 06/2019–07/2019 and found the differences to be minor at EGRIP and generally below $1\,\mathrm{W\,m^{-2}}$ in the accumulation zone of the GrIS. Given the insensitivity of the LHF to the simulated effects of blowing snow at EGRIP, we consider the effect of blowing snow on the humidity flux simulation in these four summers to have a limited impact on our results. However, it should be noted that the presented correction function in this study should not, without further evaluation, be applied to MAR simulations with blowing snow impacts included.

Our study documents large uncertainties in the SMB contribution of the humidity flux, as even the net humidity flux direction (mass loss or gain) switches after correcting the humidity fluxes. On multi-annual time scales, even small systematic errors in the humidity flux simulation have large impacts on the simulation of the net SMB. In MAR, the LHF has a correlation radius ($R \geq 0.5$) of 450 km, indicating that the results of this study are also valid for wider areas of the accumulation zone. In the present climate, our simulations suggest that the sublimation and vapor deposition cancel each other out to a large degree.

As a consequence, the net current sublimation contribution to the total annual SMB is small. However, the balance between sublimation and vapor deposition might shift in a warming climate. Simulations by Cullen et al. (2014) for the accumulation zone of the GrIS suggest that the domination of sublimation over vapor deposition will increase in a warmer climate. As a consequence, the net contribution of the humidity flux to the SMB would increase and could gain importance in the accelerating mass loss of the GrIS. We show that systematic errors in the humidity flux change MAR's simulation of the SMB in the long

term. Thus, detailed uncertainty studies concerning the humidity flux need to accompany simulations over long time periods, such as to estimate the accumulation zone extent or in ice sheet stability climate predictions.

## 4   Conclusions

This study aims to provide a reliable estimate of the humidity flux contribution to the SMB in the accumulation zone of the GrIS. This is achieved by combining simulations of the regional climate model MAR with a new dataset of high-resolution

EC flux measurements. We evaluated the LHF in the MAR model simulation with four years of continuous meteorological observations at the EGRIP location. In summer, MAR reproduces the magnitude, distribution, and diurnal cycle remarkably well but has a bias in the LHF towards a negative (depositional) flux of -1.3 W m$^{-2}$. In winter, MAR consistently overestimates the magnitude of the LHF. Both the summer bias and the winter overestimation for the LHF are caused by systematic errors in $\Delta q$. The humidity gradient depends non-linearly on the surface and 2 m air temperature, as well as the resulting gradient. And

the LHF depends linearly on the humidity gradient (Eq. 1). We, therefore, proposed a simple linear correction function for the simulated LHF based on the surface saturation specific humidity in MAR. Via the saturation specific humidity, the correction accounts for the non-linear impact of the temperature on the humidity gradient and, thus, the LHF. Note that the correction is applied offline and has no effect on the simulation itself. After correcting the systematic errors in the model data, the net humidity flux is estimated to account for about -1 % of the total SMB compared to +5.6 % for the uncorrected simulation.

However, the contribution of the humidity flux to the SMB shows large seasonal variations. During summer, the net humidity flux accounts for -32 % of the total SMB, 31 % of the total mass gain is sublimated again, and vapor deposition accounts for 9.6 % of the total mass gain. Further research is necessary to address and correct the temperature biases in the MAR model to reduce the humidity flux uncertainties of the simulation. Our results thus demonstrate that the humidity flux plays a major role in the composition of the summer SMB.


     Despite the relatively small value of the contribution from the humidity flux to the SMB on intra-annual time-scales, snow parameters used as climate proxies, such as snow structure, impurities, and water stable isotopes, can be influenced by subli-

mation and vapor deposition even on such short time-scales. Recent research has documented (Hughes et al. (2021), Wahl et al. (2022)) that sublimation has the potential to overwrite the initial precipitated climate signal in the isotopic composition. Being

able to simulate the humidity flux accurately throughout the year is critical for interpreting the proxy climate signal. This study provides a robust and simple way to correct systematic, temperature-driven errors in the humidity flux simulation in the polar regional climate model MAR. The achieved accuracy of the corrected humidity flux opens the opportunity to use MAR for future investigations of how the atmosphere-snow humidity exchange influences the surface snow.

*Data availability.* The PROMICE AWS product is available at https://doi.org/10.22008/promice/data/aws (Fausto et al., 2019). The EC water

vapor flux data set is available in PANGEA (Steen-Larsen et al., 2022). The MAR simulations are available on Zenodo (Dietrich, 2023).

**Appendix A**

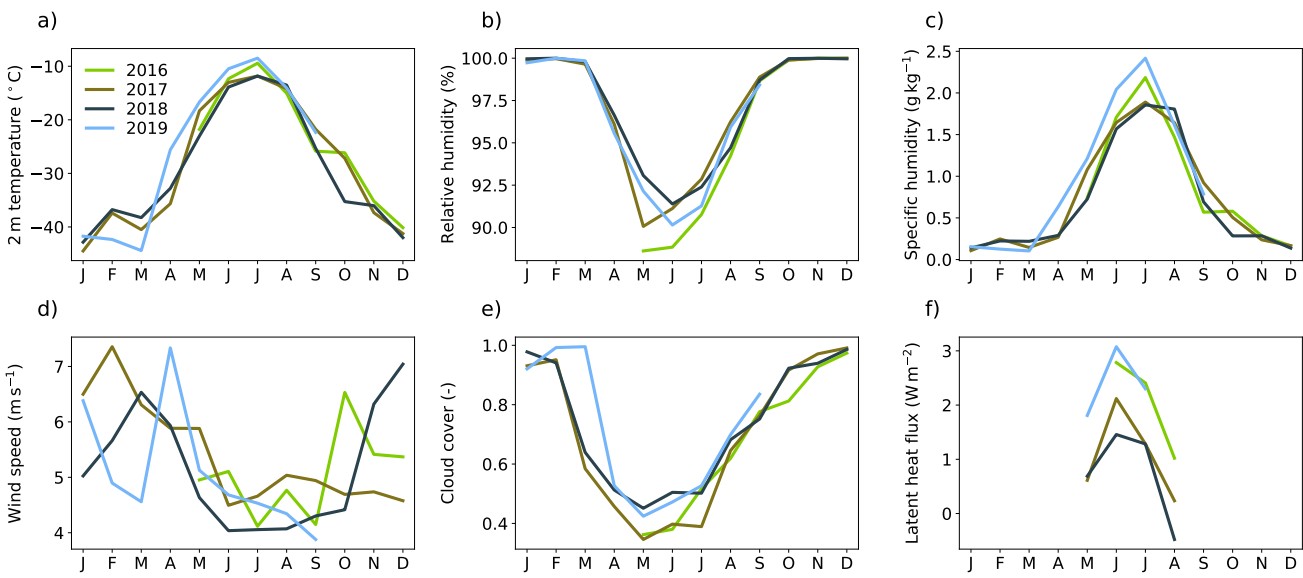

**Figure A1.** Monthly mean seasonal cycles of different meteorological variables (a-e) for the individual years in the observational period (2016-2019). All meteorological data is obtained from the AWS except for the latent heat flux that is observed with an EC system (f).

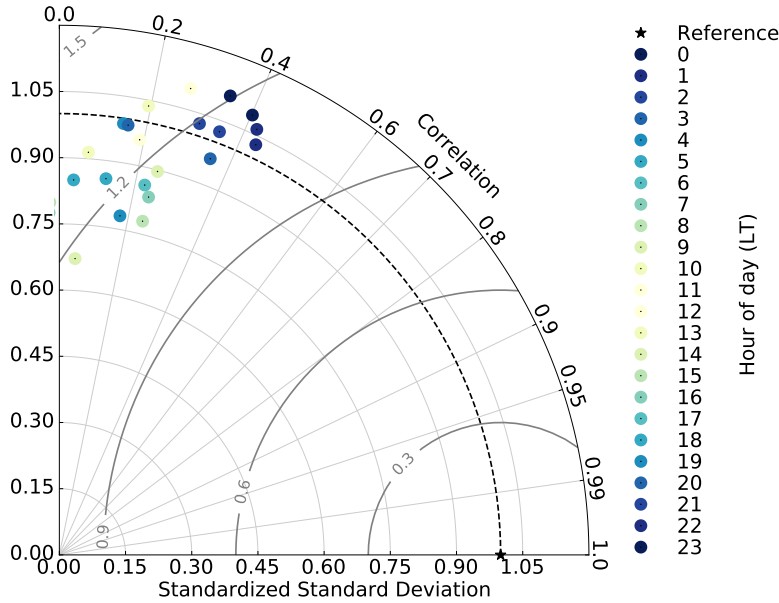

**Figure A2.** Taylor diagram of the summer latent heat flux averaged for every individual hour of the day. The reference is the EC LHF measurements for all four summers. Dark colors correspond to night hours and bright colors to day hours. The gray circles correspond to the relative error. LT is UTC-2.

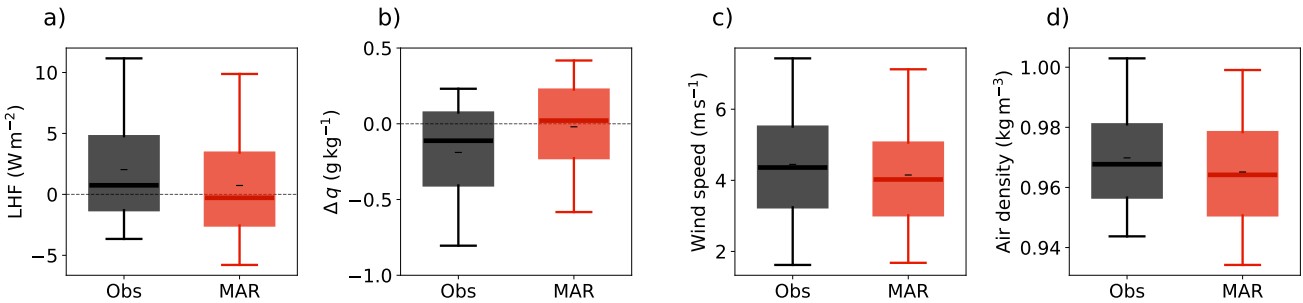

**Figure A3.** Distribution of the latent heat flux (a), specific humidity gradient ($\Delta q$, b), wind speed (c), and air density (d) for all summers (June, July) in 2016–2019 from observations (Obs, black) and (MAR, red). Black dashes (-) denote the mean, thick lines denote the median, boxes denote the 25th–75th percentile, and whiskers denote the 5th–95th percentile.

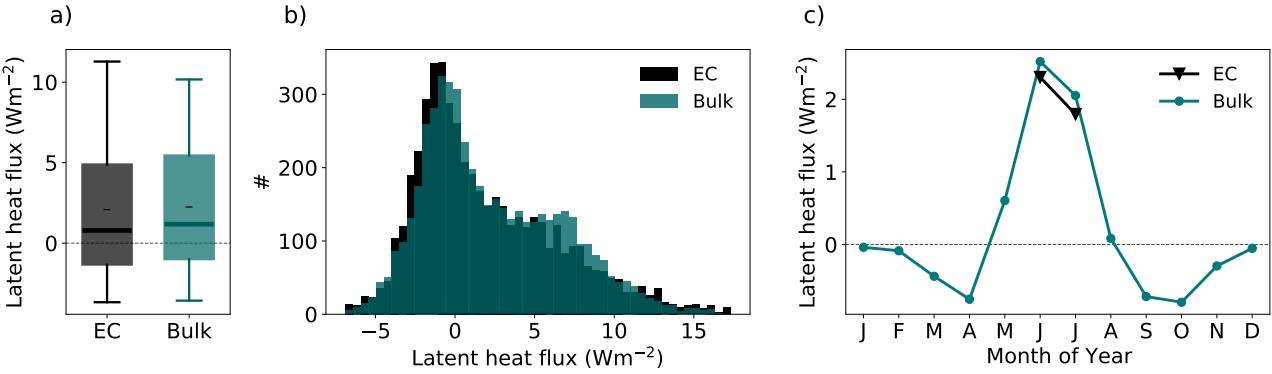

**Figure A4.** Comparison of the eddy-resolving EC LHF measurements (black) to the bulk calculation of the LHF from meteorological observations (turquoise) using a constant roughness length of $1 \cdot 10^{-5}$m in distribution (left, middle), and monthly mean seasonal cycle (right) for all summer months (June, July) in 2016–2019. In (a), black dashes (-) denote the mean, thick lines denote the median, boxes denote the 25th–75th percentile, and whiskers denote the 5th–95th percentile.

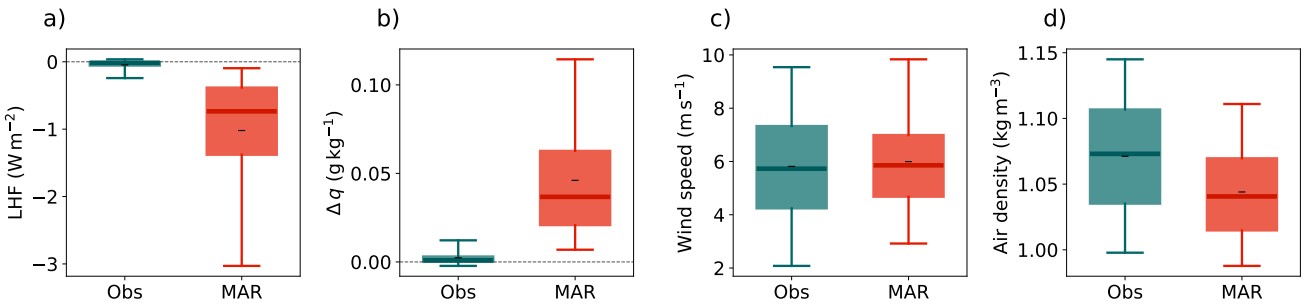

**Figure A5.** Like Fig. A3, but for all winters (December, January).

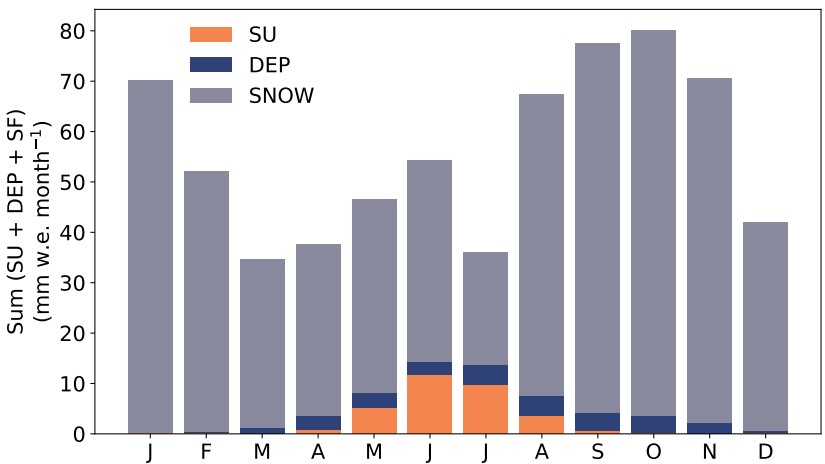

**Figure A6.** Components of the SMB, sublimation (SU, orange), vapor deposition (DEP, blue), and snowfall (SNOW, grey), at the EGRIP site simulated with MAR for all months of the year in 2016–2019. The correction function (Sec. 3.2) is applied to the humidity flux.

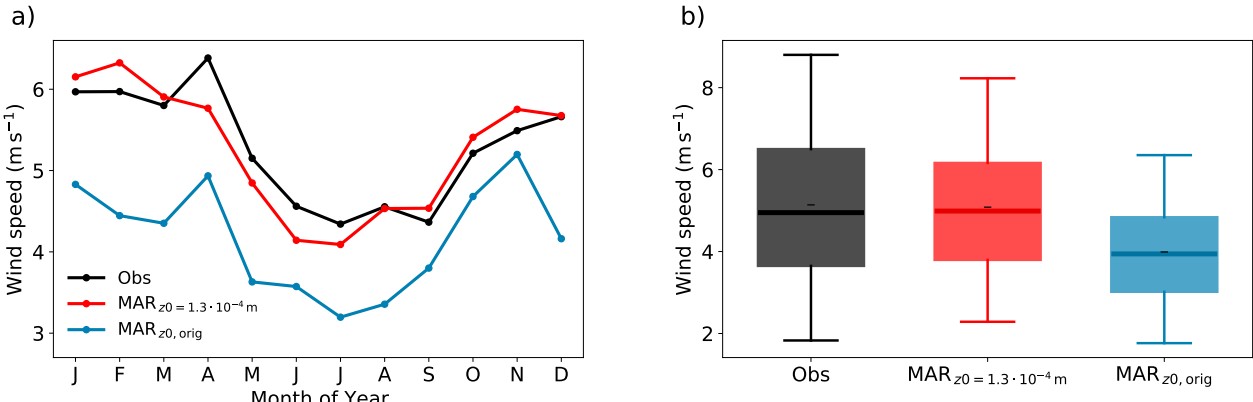

**Figure A7.** Sesonal cycle (a) and distribution (b) of the wind speed from observations (black, Obs), and in MAR using a fixed roughness length of $z_0 = 1.3 \cdot 10^{-4}$ m (red, $MAR_{z0=1.3 \cdot 10^{-4} m}$) and using the implemented roughness length parameterisation (blue, $MAR_{z0, orig}$). In (b), black dashes (-) denote the mean, thick lines denote the median, boxes denote the 25th–75th percentile, and whiskers denote the 5th–95th percentile.

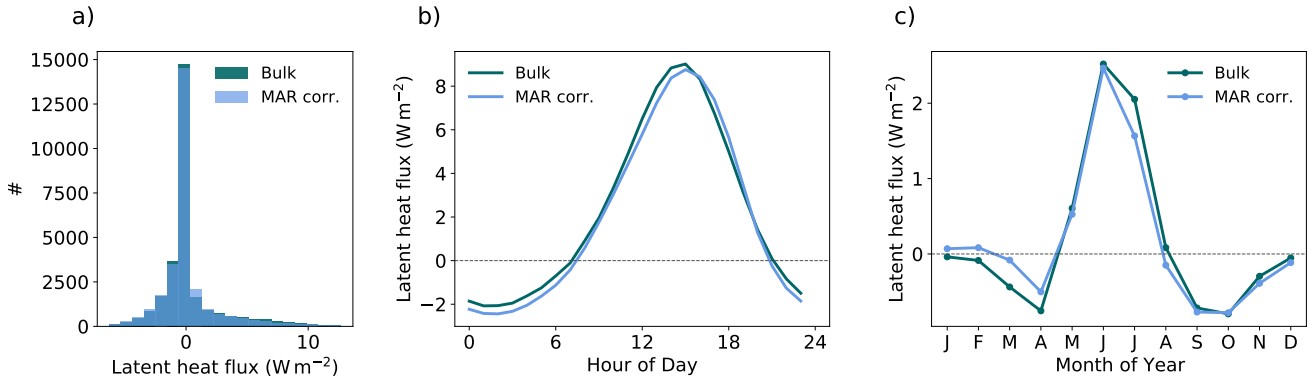

**Figure A8.** (a) Hourly data distribution in the observational period (05/2016–08/2019), (b) diurnal cycle in summer (June, July) and (c) monthly mean seasonal cycle of the bulk estimated latent heat flux from meteorological observations (turqouise, Bulk) and the corrected simulated latent heat flux in MAR (blue, MAR corr.).

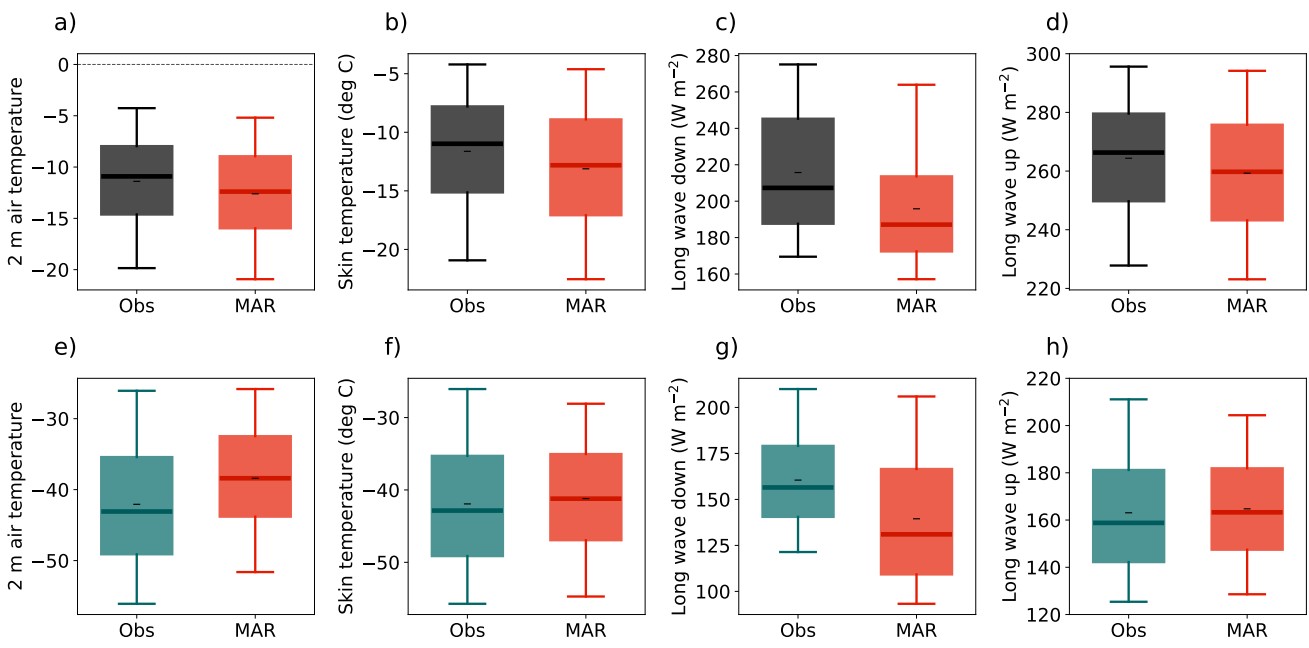

**Figure A9.** Distribution of the 2 m air tempertature (a,e), skin temperature (b, f), downward longwave radiation (c,g), and upward longwave radiation (d,h) in all summers (June & July, a-d) and winters (December & January, e-h) in 2016–2019 from observations (Obs, black) and (MAR, red). Black dashes (-) denote the mean, thick lines denote the median, boxes denote the 25th–75th percentile, and whiskers denote the 5th–95th percentile.

**Table A1.** Instrument uncertainties for the PROMICE AWS given in Fausto et al. (2021).

| Parameter | Instrument | Manufacturer | Model | Accuracy (unit) |
|---|---|---|---|---|
| Pressure | Barometer | Campbell Scientific | CS100/Setra 278 | $\pm 2.0$ (hPa) |
| Temperature | Thermometer, aspirated | Rotronic in Rotronic assembly | MP100H-4-1-03-00-10DIN | $\pm 0.1$ (K) |
| Relative humidity | Hygro-/thermometer, aspirated | Rotronic in Rotronic assembly | HygroClip HC2 or HC2-S3 | $\pm 0.8$ (% RH) |
| Long wave radiation | Radiometer | Kipp & Zonen | CNR1 or CNR4 | $\pm 10$ (%) |
| Wind speed | Propeller anemometer | R.M. Young | 05103-5 | $\pm 0.2$ (m s$^{-1}$) or 1 (%) of reading |

```python
 1: def corr_lhf(lhf, qs, fac=1.3, time=time_vec(), timesteps_in_year=8766):
 2:     """ lhf: simulated latent heat flux (LHF) from MAR
 3:         qs: saturation specific humidity corresponding to simulate surface temperature
 4:         fac: Average bias of simulated LHF compared to eddy covariance measurements in all summers
         2016-2019
 5:         time: time vector in hourly resolution
 6:         timesteps_in_year: average number of timesteps per year in simulation
 7:     """
 8:     x_orig=np.arange(12) # 12 months
 9:     x_new=np.linspace(0,11,timesteps_in_year) # 12 months equally split into model temporal
         resolution
10:     qs=seas([qs], time=time)[0] # seasonal average of qs
11:     rqs=((1/qs)-np.nanmax(1/qs)) # inverse of qs minus seasonal maximum of the inverse to normalize
         to 1 on summer
12:     m=1/np.mean(rqs[5:7])*np.tile(intp(rqs, x_orig, x_new),int(np.ceil(len(lhf)/timesteps_in_year)))
         [:len(lhf)]
13:     b=fac/np.mean(qs[5:7])*np.tile(intp(qs, x_orig, x_new),int(np.ceil(len(lhf)/timesteps_in_year)))
         [:len(lhf)]
14:     return m*lhf+b
```

**Listing 1.** Python code for correction function.

*Author contributions.* HCSL and LJD conceived the study. XF and LJD ran the MAR model simulations. SW and HC obtained the EC observational dataset. LJD did the formal analysis with contributions from HCSL and SW. Investigations were done by LJD with contributions from all co-authors. LJD wrote the manuscript with contributions of AKF and HCSL. Reviews and edits are made by all co-authors. Visualisation is made by LJD. The study was supervised by HSCL and AKF. HCSL acquired funding for this study and administrated the project.

*Competing interests.* At least one of the (co-)authors is a member of the editorial board of *The Cryosphere*.

*Disclaimer.* TEXT

*Acknowledgements.* This paper has received funding from the European Research Council (ERC) under the European Union's Horizon 2020 research and innovation program: Starting Grant SNOWISO (grant agreement no. 759526). AWS data from the Programme for Monitoring of the Greenland Ice Sheet (PROMICE) and the Greenland Analogue Project (GAP) were provided by the Geological Survey of Denmark and Greenland (GEUS) at https://www.promice.org. EGRIP is directed and organized by the Centre for Ice and Climate at the Niels Bohr

Institute, University of Copenhagen. It is supported by funding agencies and institutions in Denmark (A. P. Møller Foundation, University

of Copenhagen), USA (US National Science Foundation, Office of Polar Programs), Germany (Alfred Wegener Institute, Helmholtz Centre

for Polar and Marine Research), Japan (National Institute of Polar Research and Arctic Challenge for Sustainability), Norway (University

of Bergen and Trond Mohn Foundation), Switzerland (Swiss National Science Foundation), France (French Polar Institute Paul-Emile Vic-

tor, Institute for Geosciences and Environmental research), Canada (University of Manitoba) and China (Chinese Academy of Sciences

and Beijing Normal University). The simulations were performed on resources provided by Sigma2 - the National Infrastructure for High

Performance Computing and Data Storage in Norway. We thank Mike Town for valuable input.

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
