# Peer review of "On the importance of the humidity flux for the surface mass balance in the accumulation zone of the Greenland Ice Sheet"

_The Cryosphere, 2022_

## Referee Comment (RC2)

Review of manuscript tc-2022-260 "On the importance of the humidity flux for the surface mass balance in the accumulation zone of the Greenland Ice Sheet" by Dietrich et al

**General comments**

The paper presents an analysis of humidity flux for a site in the accumulation of the Greenland Ice Sheet. Data from summertime direct eddy-covariance (EC) observations, AWS-based bulk-aerodynamic model estimates and output from a regional climate model (RCM) simulation are presented and discussed. The bulk estimates are found to give good agreement with summertime EC data. The RCM simulations are shown to have seasonally dependent biases. Correction functions are derived, and the corrected humidity fluxes used to comment on the importance of humidity fluxes for the surface mass balance.

The topic should be of high interest to readers of *TC* and the paper presents a useful combination of new observations and modelling results relevant to the topic. A mix of established and new methods are used, most of which are well described and suitable. The figures are very well made, and the text is concise and mostly clear. Several useful results are presented and discussed. Relevant previous research is cited and discussed.

However, further discussion and additional results are needed to support the stated conclusions:

While the idea of a scale and offset correction appears to be a useful way to scale the RCM simulations, I would argue the results show that MAR does not capture the all the relevant processes well, particularly in the wintertime. This may limit the applicability of MAR results in other areas, and further discussion is needed.

More analysis and discussion of the reasons for the biases in MAR humidity fluxes is needed. This should include the interaction of LW radiation, katabatic winds, blowing snow, and sub-surface temperatures on the profile of temperature and humidity in the lower atmosphere.

I am concerned about the exclusion of blowing snow from the MAR simulations, particularly as it has been shown elsewhere to greatly affect the surface humidity flux. The choices made here need further justification, and at least some sensitivity analysis within MAR is needed to show the effect of including blowing snow or not.

Further considerations of the limitations of the bulk method for estimating accurate wintertime humidity fluxes, especially in blowing snow conditions.

With these additional results and discussion, the paper should make a useful contribution to *The Cryopshere.*

**Specific comments**

Title – given the above uncertainties in both the bulk and RCM simulations, a more targeted title is warranted e.g. "Correcting regional climate model estimates of humidity flux in the accumulation zone of the Greenland Ice Sheet with in-situ observations."

7 – It would be more consistent to present the humidity flux here in mass units (i.e. mm w.eq.)

97 – the explanation given for why blowing snow in MAR was turned off is unclear. Please expand.

127 – please provide further detail of how the setting used to process the EC data. In particular whether any data were excluded or gap filled from the summer records.

130 – please either provide additional details and statistics of the EC comparison, or exclude this comment

162 – earlier you comment that 2019 was warmer and more humid than other years – how does this affect its representativeness.

165 – please clarify the direction of a 'negative' humidity gradient in the text – the definition above only states it is the difference between 2m and surface.

175 – please clarify it is the 'MAR simulated LHF' (as the bulk estimates are a simulation of sorts).

198 (also 212 and elsewhere) – it would be clearer to talk about 'offset' and 'scale' biases here – where the summer is primarily an offset bias, whereas in the winter and offset in the magnitude of the flux also produces a bias (difference between means of observations and simulations)

214 – how is 0.1 a 'zero-bias'? please revise.

Figure 8 – please provide further detail on how the seasonal curves for the factors $m$ and $b$ are derived?

224 – there appears to be an inconsistency in how the annual and summer humidity flux contributions are reported as the text differs (i.e. "In the corrected simulation 5.1% [4% to 6 %] of the annual mass gain (snowfall + deposition) sublimates again," vs "During summer, the amount of sublimated mass corresponds to 31% [26% to 34 %] of the total mass gain".  If these are calculated in the same way, consider using the same language for each to avoid confusion.

230 – "Our evaluation shows that MAR captures all relevant processes driving the humidity flux and captures the distribution of the LHF remarkably well (Fig. A8)" I would disagree with this statement. The existence of the biases in the uncorrected simulations highlight that important processes are likely missing from the simulation. But that with correction can be corrected to represent the distribution of humidity flux. Please revise.

237 – "bias of -1.3Wm−2" it is awkward to introduce an energy term here without context. A mass unit would be better suited.

242-246 – the distribution of biases in surface and air temperature, and incoming LW needs to be shown here.

248 – "systematic error in the LHF is not a bias" as for line 198, the error is  a bias, just a different sort of bias – please revise.

255-262 – please show the distribution of biases of variables related to the humidity gradient. These are key elements of evaluation and deserve further description and discussion.

291 – how do we know that blowing snow events are rare? The average wind speed is high, and with a dry snow surface, it is likely that blowing snow events are frequent.

**Editorial comments**

249 – "RMSE = 0.73Wm−2" – correct units

---

## Author Comment (AC1)

*Responses to the reviewer are given in* **green***.*

*General Comments:*

*In an effort to increase the accuracy of a regional climate model (MAR), the authors focus on one term in the energy flux of the snow surface over the Greenland Ice Sheet. The method inherent to MAR, a bulk flux estimate, is widely known to be inaccurate. The authors compare fluxes estimated by this bulk approach to fluxes estimated by an eddy-covariance system at the EGRIP drilling site. The authors then use a linear correction to make the bulk estimates more closely match the eddy-covariance estimates. This is an incredibly ambitious problem to approach, and as such the results need to be rigorously supported.*

**We generally agree with the reviewer's summary. There might be a small misunderstanding as it is not correct that we tried to match EC and bulk estimates by correcting the bulk estimates from the AWS. Instead, we post-correct the model simulation that is based on a bulk calculation using the observations, which consist of a EC dataset in summer and bulk estimates to cover fall to spring seasons. We understand that the origin of this confusion is probably based on some imprecise formulations in our article, and we updated the text to make that clearer.**

*This linear rescaling works relatively well given its simplicity. Such a simple correction is necessary to maintain any sort of numerical economy for a regional climate model. However, the authors opt to rescale the output of an LHF estimation, not the constituent physical variables, which themselves influence other terms in the energy balance. Given this, the way in which the results are presented needs to be very precise. Further comments on my concerns and issues this might create are discussed below.*

**We agree that post-correcting simulation output does not improve the simulation itself and ignores the causes of simulation errors. In this study, we don't aim to correct the simulation to provide better model output from MAR, but we rather want to show the mismatch between the model simulation and observations, as well as the potential long-term error of such a mismatch in the simulated SMB. We updated several parts of the article to clarify and stress the uncertainties that come with post-correcting the data instead of addressing the causes of the errors (see responses to detailed comments below).**

*Furthermore, there is an underlying implication that the eddy covariance method can be used to actually measure the vapor transport away from the snow surface. This is not actually the case and there is 25 years of publication in Boundary Layer Meteorology attempting to understand and better estimate vapor fluxes using Reynolds-decomposition-based methods for both stable and unstable atmospheres. To date, the issues and errors associated with EC over snow are ample, even in relatively homogenous terrain. When reading this manuscript, the feeling that the authors give is that EC calculations can actually be used as a ground truth with which to improve MAR. This is, unfortunately, not scientifically accurate. That being said, the conclusions of the manuscript are not without their own merit. Modifications to the language and conclusions that reflects the underlying lack of an actual "ground-truth" would make this more suitable for publication. Reconsider using "observed LHF" (or justify) as the EC approach is itself also only an estimate.*
*Specifically, I think the introduction should be improved by being both more precise about the studies they are referencing as benchmarks and including more references to show general issues in using turbulent fluxes in stable boundary layer and over snow-covered terrain to represent water vapor transport (as mentioned above). The authors are suggesting that there are errors in previous experiments and models that represent "humidity flux", and provide a general view that EC estimates are better than bulk estimates. While few people (if any) would argue bulk estimates are more physically accurate, there are still significant variations between EC studies, especially relating to ice sheet/snow surface conditions and the instruments that are used to measure water vapor.*

**We agree that the discussion on the uncertainties of the EC method above snow surfaces has been deficient and that more context is needed to classify the quality of the EC dataset. We updated the introduction, added the underlying assumptions of the EC method, and provided more information on the referenced sources (see responses to detailed comments below). Further, we improved the discussion of the uncertainties in the EC dataset used in this study. We addressed the language throughout the entire text to prevent the reader from understanding the EC measurements as "ground truth" (see responses to detailed**

comments below).

Additional information for the Reviewer: The EC dataset was quality-checked in a cross-validation of three independent EC systems at EGRIP in 2019. The conclusion from this comparison was that the EC data obtained in the fours summers at EGRIP has a comparably high quality. The agreement between the three EC systems was excellent with deviations usually below ±10 % with a correlation of the 30-minute data of well above 0.9 for a continuous measurement period of two months. This indicates that the turbulence at EGRIP during the summer is well-developed and sufficient for EC measurements, as well as a sufficient heterogeneity of the flow. The EC system comparison at EGRIP was conducted by a Master's student and will be turned into a publication. Therefore, we cannot publicly share this data, but both reviewers will get access to the relevant figures via the editor.

*Fundamentally, I am concerned that there is also no consideration of blowing snow processes in this study, which have been shown to be responsible for sublimating up to 50% of seasonal SWE in some environments. Presumably in the cold, dry interior of Greenland significant latent heat will be exchanged by this specific sublimation contribution. In order to represent an accurate surface energy balance, without specifically addressing blowing snow, both the bulk approach and EC estimates would be influenced by blowing snow sublimation in their calculations (it's actually happening and affecting the humidity), and potentially other terms that would impact surface mass balance. It appears the authors wish to not specifically address this source of latent heat flux, and wrap its influence up into the corrected bulk estimates. If this is their agenda, it should be specifically addressed and justified why this is a scientifically sound approach.*

We completely agree that the role of blowing snow as part of the surface mass and energy balance needs to be quantified for the Greenland Ice Sheet. While some few evaluations of the blowing snow module in MAR have been made for the Antarctic Ice Sheet, it has not yet been evaluated against any observations for the Greenland Ice Sheet, neither for the surface mass balance nor for the surface energy balance. Observations at EGRIP used do not provide information on the amount and occurrence of blowing snow. While for the summer seasons 2016–2019 we can report from experience that blowing snow events were very rare, we have no information about fall to spring.

We decided to keep the blowing snow module turned off in MAR for two main reasons: 1.) To keep simplicity and to avoid the potential canceling of two error sources (i.e. temperature biases and systematic errors in the blowing snow simulation), and 2.) to exclude local spacial variations introduced by the blowing snow module from the analysis that can not be expanded to the surroundings of EGRIP due to missing observations. By this approach, the mismatch between the model and the observations can be directly accounted for all error sources, the uncertainty of the observations, the known temperature biases in MAR as well as the neglect of blowing snow in the simulation. Since post-correcting the LHF solely based on the temperature(-gradient) biases in MAR led to a strong improvement of the match between observations and the simulation, this could indicate that blowing snow plays a smaller role at EGRIP than for many Antarctic locations. However, this is speculation, and more observations are needed to benchmark the role of blowing snow in EGRIP and to evaluate MAR. We hope to set up a system to observe blowing snow in winter 2024/2025 at EGRIP to provide more insights in a future study.

We acknowledge that the discussion of neglecting blowing snow in the simulation has not been comprehensive enough in the original version of this study and we now provide a clearer discussion of the potential impacts on our key results, the correction, and the limits of the applicability of the post-corrected data (ll.393-356).

*I was also a bit surprised in the results section when the authors deviate from their proposed objective and instead use lower-frequency observations and a bulk approach for part of the year to compare with the bulk approach in MAR. The justification of this appears to be that they can make bulk estimates from observations close to EC (in linear terms), and thus it's good to then get close enough to the bulk observations. There's a compounding error here that has not been commented on, nor has the extensive use of linear statistics and regressions for nonlinear functions.*

*As well, this EC data referenced does not actually comprise the majority of the data that is used to calculate the humidity flux corrections. It seems that the authors may actually be overstating its influence on the model correction.*

**We agree that relying on bulk estimates for autumn to spring for the post-correction is a compromise that was mandatory to investigate the long-term annual SMB. To our knowledge, there are currently no multi-year EC measurements available for the GrIS that are validated to other LHF measurements. The aim of this study was to post-correct the simulated LHF in MAR to match it as close as possible to the observations. Including the bulk estimates for fall to spring not only allow us to investigate the role of the LHF in the total SMB using MAR, but further to estimate the potential impact of systematic errors in the LHF simulation on the SMB. We highlight the uncertainties of both the EC and bulk data now in the discussion.**

*I like the idea of the manuscript, but suggest major revisions prior to publication.*

**We thank the reviewer for the thorough, very constructive, and clear feedback. We have now strived to outline the uncertainties and caveats of our methods more clearly and added more information on the EC dataset and the impact of blowing snow on the MAR simulations. We respond to the detailed comments below.**

*Specific Comments:*

*L25 Reconsider using deposition as this may be confused with deposition of precipitation*

**We use the term "vapor deposition" now instead.**

*L25-27: Are you considering sublimation from blowing snow as well? Sublimation rates of snow particles in transport are much greater than that found from the snow surface. Blowing snow sublimation has been shown to play a significant role in surface mass balance in a wide range of alpine, arctic, Antarctic and other cold regions, and has even been connected to seeding of clouds. Please justify or explain early on.*

**Yes, sublimation from blowing snow is considered here as well. We updated the sentence and mention this aspect now (ll. 28).**

*L42-43: I would suggest against referring to eddy-covariance as a "direct measurement." There are significant assumptions that need to be satisfied for the eddy-covariance to approximate turbulent fluxes, which themselves are a simplification of the actual vapor flux at a given height in the atmosphere. These assumptions and errors need to be discussed.*

**We agree with the reviewer's concern that the term "direct measurement" might be misleading. We, therefore, exchanged the term "direct" for "eddy-resolving" throughout the article.**

*L43-44: What is your definition of "humidity flux" that you are using if you are not referring to latent heat flux? I think this needs to be precisely defined as you use both terms, but they do not appear to be interchangeable for you.*

**The reviewer is correct that the terms "humidity flux" and "latent heat flux" are interchangeable as we use them, and the differentiation serves only to stress either the mass flux or energy flux associated with the LHF. We added a definition in ll. 45-46.**

*L44-47: Again, are these estimates taking into account a mobile snow surface or not? For example, blowing snow sublimation specifically has been attributed to removing 200 mm yr^-1 of SWE on the Antarctic ice sheet (Lenearts et al., 2012). Please clarify the two different methods in this study. Both bulk approaches?*

**Yes, since these estimates are based on a two level gradient bulk method, and a one level bulk method. Since the humidity gradient is affected by sublimation of drift snow as well, this is included in their estimates. The lines are updated with the additional information (ll. 46-49)**

*L53-54: You mention limited accuracy of Monin-Obukhov similarity-based approximation, but do not mention any of the limitations to EC calculations in the manuscript (transience, stationarity, gravity waves, etc.). This should be remedied.*

**Yes, we completely agree with the reviewer and list the assumptions and limitations in the introduction (ll. 72-74) and discuss the appropriateness of these assumptions at EGRIP (ll. 280-288).**

*L61-62: Is this because blowing snow sublimation saturated their modeled surface layer and effectively turned off the sublimation model? This is a common issue with sublimation modeling over snow when there is no process built in to account for advection to/away from the saturated surface layer.*

**Here, we refer to the underestimation of the LHF when they apply Bbulk parameterization to their model simulations, and not the underestimation of the LHF of the model. Thus the chosen parameterisation scheme has no impact on the humidity level in the simulation.**

*L65-66: A statement should probably be made that the roughness length itself is not a physical thing to be obtained, rather a term used to describe the influence of turbulence on an assumed log-linear profile. That is, a correction to a simplified model that is trying to represent nonlinear processes.*

**We agree with the reviewer that these lines needed clarification and added a statement on the roughness length (ll. 68-69)**

*L66-67: This language is a bit concerning as you have not as of yet shown that the "direct EC method" actually is capable of measuring humidity of water vapor fluxes in this environment. Either rephrase this, or reference literature that shows the accuracy of measuring humidity fluxes with EC. I imagine the latter is not an option as people are still unable to close energy balance models with measured variables (e.g. Helgason and Pomeroy, 2012a,b; Harder et al., 2017). There is always a question in boundary layer studies of, "are you getting the right answer and for the right reason?" It would perhaps be sufficient to show that studies attempting to close the surface energy balance over snow get small discrepancies when using EC versus bulk approaches, but the language needs to be precise if there is no proof the individual flux term is being measured more precisely.*

**The relatively high confidence in our data origins from the unpublished cross-validation in 2019, mentioned above. We agree with the reviewer that setting up an energy balance as part of the cross-validation study would provide valuable insights to assess the data quality of the published dataset used in this study. We rephrased the sentence (ll. 74-76) and treat the EC dataset with much more care throughout the manuscript.**

*Furthermore, you are specifically talking about estimating EC fluxes of humidity using high temporal resolution water vapor measurements, correct? Accurate measurement would require a closed-path style water vapor measurement as snow particles would impact the signal quality, such as with a KH20. Otherwise, denoising will be required, which can take up a significant amount of a time series during storm events. I do not know of many experiments that actually use this sort of apparatus over snow, and it would significantly benefit you to list such experiments here.*

**Again, we reference to the additional material that will be given to the reviewer. A study that measured with a comparable EC system is Miller et al, 2017. We updated the references in the introduction (ll. 76-77).**

*L86: What surface processes are you considering?*

**In MAR the simulated surface processes cover snow melt, sublimation, and vapor deposition, refreezing, changes in snow optical properties, and snow texture. We updated the text with these information (ll. 95-96).**

*L96-99: I don't understand how turning off blowing snow in your model makes your findings more broadly applicable. This is a complex physical process that is local in nature, and it would be great if you could justify ignoring it*

**As mentioned above, the fact that the blowing snow module has not been evaluated against observations in Greenland despite its complexity leaves us with doubts of the reliability of this snow module. We, therefore, prefer to exclude the impact of the blowing snow module in MAR on the results but discuss the impact of this decision on our results now more prominently (ll. 393-356).**

*L123-131: This single paragraph on "Atmospheric eddy-covariance system" needs to be significantly expanded. This is essentially the backbone of the manuscript. This data is the data that all subsequent conclusions are based on. I see the data has already been published, and its investigation will not be the focus of the present work, but why was 30-minutes chosen? Was this a statistically stationary window? Were there any large-scale amplitude modulating structures inside this 30-minute threshold? Why do you then average to hourly time scales? To match MAR? How much of the data was thrown out for being noisy? Is there any bias in the data that you selected e.g. did you ignore especially high or low wind speed periods? Especially cold times? Hot times? I see 5304 data points, but there are at least 5700 hours spanning those months in those years. What happened to the rest and why is it not there? As with any model tuning study, it is important to know how representative the data is you are using for tuning and validating your tuned model. Including a figure that shows how representative these values are would be very beneficial, as well as more discussion.*

**This is very constructive feedback to improve the quality of the manuscript. We updated the paragraph with relevant information (ll. 134-149): We now provide information on the integration time which indeed has been shown to guarantee optimal stationarity and averaged to hourly data to fit the model's frequency. The filter that was applied to the published dataset is a simple noise filter by applying a cut-off threshold of fluxes <-20 W/m2 and >40 W/m2 that is now given in the manuscript but no filtering based on the wind speed or temperatures is applied. The representativeness of the data is shown in the supplement material that is provided for the reviewer but not published yet.**

*As well, this EC data does not actually comprise the majority of the data that is used to calculate the humidity flux corrections, and the measurements used for the bulk calculation are discussed even less.*

**The caveat that despite its uncertainties in estimating the turbulent humidity flux in Greenland, we use bulk estimates for our correction function is now pointed out (ll. 288-291), and the uncertainties of relevant sensors of the AWS are given in the Appendix.**

*L129: Introduce this equivalence earlier when discussing LHF, EC, and humidity flux.*

**Moved into the introduction (l. 71)**

*L129-131: I am still a bit confused at this point. Where does this confidence in the EC calculations come from? I get that the EC instrumentation may have been shown to be reliable. However, if I understand correctly, you are using covariance, as calculated from fluctuating time series of your vertical wind speed and some representative of water vapor, to represent the flux of water vapor away from the ice shelf surface to the atmosphere. From this physical standpoint, how do you know you have done that correctly? As mentioned before, I assume you have not been able to perfectly close an energy balance model at this site using measured variables, correct? Or a model of snowpack evolution? As you state in the introduction "This study addresses the uncertainty in regional climate model estimates of the humidity flux contribution to the SMB in the accumulation zone of the GrIS" but is there not already uncertainty in the humidity flux contribution to the SMB from your own measured data because the SMB and EMB have not been closed?*
*Please rephrase or explain.*

**Again, we thank the reviewer for this very valid and constructive critique. We agree that since the EC dataset is an essential element for the methodology in this study, more insights into the production of this published dataset are needed. We updated the entire section and now give more information on the choice of integration time as well as on the filtering of the published EC LHF dataset. As mentioned above, the reviewer will receive information on the cross-validation in 2019 to estimate the EC data quality.**

*L137: Upstream "in" the prevailing wind direction?*

**Is corrected (ll. 155).**

*L161-162: Why is the summer of 2019 a good representative? Earlier you said that it was warmer and wetter.*

**Indeed, this is formulated too strongly. By "good representative" we were referring to the fact that the distribution of the LHF in the summer 2019 is very similar to the total distribution in all four summers. Using an example (in Fig. 3 & 6) makes it easier for the reader to understand the different types of systematic errors in winter and summer. We removed the valuation "good" and present the summer 2019 only as an example now (ll.179-181).**

*L162-163: R-values are not valid for nonlinear regression (e.g. Spies and Neumeyer, 2010). Please use a more representative statistic.*

**The reviewer is of course correct that R values are not a valid statistical measure for non-linear regressions and the authors apologize for not being clear enough in these lines: The regression is made between the same meteorological variable, one simulated and one observed. Therefore the regression is linear and the R-value refers to the correlation between, e.g. the daily wind speed in MAR and the observed daily wind speed. Thus, we argue that the R-value is the correct measure to use here. To clarify for the reader what the R-value refers to, we renamed all linear R-values in the manuscript to $R_{MAR-Obs}$ instead of R.**

*L169: Why is this direct?*

**Changed to "eddy resolving" (l. 189).**

*L170-171: Is this the PROMICE AWS? Please clarify. It seems like these bulk fluxes constitute the majority of the months where you do your fitting, so this data needs to be in your methods section as it is a significant component of your model correction.*

**The reviewer is correct that these bulk estimates are based on PROMICE AWS data and that the correction function is based on these bulk estimates for fall to spring due to a lack of EC estimates for these periods. We updated the lines to clarify the origin of the meteorological data (ll. 189-191).**

*L171-173: What observed roughness length are you referring to? Why would roughness in summer and winter be the same? You have different turbulent mechanisms. Furthermore, what are you comparing to get best agreement with EC of LHF in summer? Did you run the bulk estimate year round? I think it would help to refer to the three LHF calculations as LHF_Mar, LHF_EC, LHG_AWS or something similar. This is getting a bit hard to follow.*

**The observed roughness length is based on the EC measurements in summer 2019. It is correct, that the chosen roughness length provides the best agreement between the bulk estimates and the direct EC observations of the LHF in all summers 2016-2019. The text is updated with this information (ll. 192-194).**

*L174: Nonlinear use of R.*

**As for comment on L162-163. R is renamed by $R_{MAR-Obs}$.**

*177-178: Linear correlation again.*

**As for comment on L162-163. R is renamed by $R_{MAR-Obs}$.**

*L179-180: Please rephrase: "twelve values out of four years per season"*

**Updated to: Note that these seasonal correlations are only based on four years, resulting in a total of 12 different monthly values out of each season (ll. 200-201).**

*L183: I am a bit confused now how you can come up with a seasonal error when the method you are using to generate "observed" LHF is different in winter and summer. I don't think any seasonal cycle would be meaningful since your observation source is significantly changing.*

**As mentioned above, the reviewer is correct in pointing out that basing the correction function on the combination of two observational datasets that are based on two completely different observational methods is a weakness in our methods. Figure 5, therefore, shows both the seasonal cycle of the bulk estimates (dark green) as well as the monthly averages of the EC estimates (black). In the overlapping period (May to August), the monthly values of both observational methods are very similar, and, thus, MARs offset in the monthly means during May to August is of similar magnitude compared to both datasets. The highlighted sentence in l. 183 was indeed confusing, and we updated it for clarification (ll. 205-206).**

*L190: Again, why good?*

**The distribution of the LHF in winter 2018/2019 is similar to the total distribution of all the LHF in winter months, which is why we chose this winter as an example. We updated l.190 and removed the valuation "good" but only refer to this winter as an example now (ll. 212-214).**

*Figure 3: Please clarify the source of these observations. All EC here, or are you relying on the AWS as well?*

**The LHF is based on the EC measurements, specific humidity gradients, wind speed, and air density are obtained from the AWS. The caption of Figure 3 is updated.**

*Figure 7: Please improve the explanation of this figure. What is bulk? I thought you were using EC. Is this winter only?*

**The caption of Figure 7 is updated. The shown observations are bulk estimates using meteorological observations from the PROMICE AWS (dark green, bulk). The figure shows the cumulative sum of the bulk estimates from the AWS of the scope of 41 months. Because the EC data is only available in summer months, the bulk estimate appears more suitable for an analysis of the multi-year cumulative sum.**

*L203-204: If these correction terms are functions of time, please clarify that in your equation.*

**The parameters are based on the monthly means of the surface saturation specific humidity: f(LHF) = m($q_{s,sat}^{-1}$) LHF_MAR + b($q_{s,sat}$), and, thus, vary in time. The text is updated (ll. 272-273)**

*L209: what do you mean it's based on q_s,sat? Is it a function of q, or is that just something that influences the bias you are correcting for? Likewise for your explanation of m.*

**The reviewer is correct in assuming that b is a function of the monthly averaged surface saturation specific humidity and m a function of its inverse. We post-correct the LHF by applying this simple function in order to estimate both the potential impact of the systematic errors in the MAR simulation and the role that humidity fluxes play in the SMB at EGRIP. We give the exact calculation now in the text (ll. 231-233) and attached the python code in the Appendix.**

*L211-214: Can you explicitly write out the functions that you used, or the periodic functions that you fit for m and b? Right now, I don't think people could recreate your results. From figure 8 it looks like you did a linear fitting each month? Can you clarify how you did this?*

**See reply for comment on L209.**

*L219-227: If I understand your process correctly, you are taking LHF output from MAR, you found a linear rescaling to match a combination of EC and Bulk observation data on a monthly timescale, adjusted your MAR output accordingly, and then quantified how the adjusted data (now at an hourly timescale) adds up when combined with non-adjusted MAR data to generate an SMB. Is that correct? Given that the other terms in MAR*

*are also likely incorrect, but are as of yet not corrected, how can you make any conclusions about what is actually happening to a given quantity of snow that falls on the ice sheet? It looks like you have kind of brute forced sublimation into your model (Figure 7a) and then conclude how much sublimation is then happening. But there are nonlinear feedbacks between other parts of the surface energy balance that you have no corrected, or adjusted. Why would these changes in sublimation rates not be countered by some other as of yet uncorrected flux term should you do something like include the change in temperature that will result from these increases in sublimation? Or even amplified? Since there is no feedback into other MAR output, I don't yet understand the impact of tuning the model for more sublimation and then reporting on the magnitude of that sublimation change. I am not trying to be cynical, and I think an explanation of the implications could really help me. Perhaps a deeper explanation in the style of Figure 6 could explain why there are no other terms in the energy balance that NEED to be corrected? How can you make a Figure 6 once you have done your correction?*

**We understand the concerns of the reviewer, and we agree that applying our correction function does not provide a new dataset of accurate LHF simulations. The aim of the study is to show the potential magnitude of the systematic errors in the current LHF simulation and their impact on the total SMB on climatic time scales, as well as to highlight the important role that the LHF plays in the accumulation zone SMB. And to serve this goal the correction function is applied to the LHF output of the MAR model. We further highlight that current surface-near temperature biases need to be addressed to accurately estimate both the SMB and SEB of the interior Ice Sheets. While we agree that the SEB would be highly impacted by such high changes in the LHF, the amount of precipitation is likely less affected by changing surface humidity fluxes. We expect the amount of precipitation to depend mainly on the atmospheric humidity (and ocean temperature) and atmospheric temperature west of the GrIS and the temperature above the GrIS at the height where the precipitation forms. Thus, a SMB estimate based on the precipitation from the uncorrected simulation is appropriate. We fully agree with the reviewer's point that such big change in the LHF would potentially affect the surface-near humidity and could thus cause a negative feedback on the LHF magnitude. However, the LHF is corrected to match the estimated fluxes from observations as the best available estimate of the true humidity flux, already including such feedback. We updated the paragraph by pointing out that these numbers are not based on a corrected simulation of a climate model to avoid overconfidence in the exact numbers of the humidity flux contribution (ll. 255-257).**

*L230: How did you come to this conclusion? Please rephrase within the constraints of the study.*

**The conclusion is based on the already striking performance of MAR to reproduce the distribution of the LHF aside from the biases. Such good performance in simulating the LHF above Greenland is not common among climate models and shows that MAR is a suitable tool for SEB investigations. We rephrased the entire paragraph within the study's constraints (ll. 261-278).**

*L258-259: This difference in temperature gradient supports my previous statement as sensible heat fluxes would also be influenced by the sort of physical change that is being imposed on the system by forcing terms in the LHF to be rescaled. In hopes of being clear about this point, q in equation (1) is obviously a function on the state of the system, thus influenced by temperature, as is u. By forcing LHF to change, you are implicitly changing the terms inside 1 that are not constants, but you are not accounting for those changes in other energy balance terms. There appears to also be a potential impact on longwave radiation terms. Why can we disregard the nearly 4 degree difference in 2 meter temperature for longwave radiation, or sensible heat, but state that it is important LHF? And how do we know accounting for ALL the relevant changes won't result in a different cumulative effect?*

**We completely agree with the reviewer's point of view. The humidity flux is commonly stated to be minor in the GrIS SMB. However, this statement is poorly constrained by observations and as such probably not correct. By post-correcting the fluxes we were able to point out the potential role of the humidity flux in both the SMB of the GrIS accumulation zone and the simulation of the snow properties, including the stable water isotopes. In addition, we stress the importance of understanding and correcting the causes of temperature biases in MAR. However, this is not an easy task and further research needs to be done. Nevertheless, the authors found it important to publish these findings on the potential long-term impacts of the existing model biases on the SMB using the well-established SMB model MAR. We now specifically highlight this caveat of our approach in ll. 267-278.**

*L278-281: This should probably be explained back in the methods section.*

**We moved this part to the methods (ll. 114-118) and added a shorter reminder of the used roughness length in ll. 334-353.**

*L285: Why does disregarding blowing snow support validity of your results? I don't understand this.*

**The statement referred to the previous sentence, stating that blowing snow can cause local mass variations in the simulation that would erroneously be accounted for as new precipitation input. Because in our analysis, we investigate the impact of humidity fluxes on the SMB for a single location of the GrIS, turning off blowing snow excludes this uncertainty of local variability due to blowing snow. We removed the statement as it is misleading and updated the paragraph to stress the relevance for our results of turning the blowing snow module off in the simulations (ll. 339-356).**

*L290: Why do you assume blowing snow events are rare and insignificant? A relatively high threshold windspeeds of 5m/s is often met in Figure 6. Sublimation rates of blowing snow have been shown to be relevant even in high accumulation alpine zones.*

**We thank the reviewer for pointing out this inaccurate formulation. In fact, information on the occurrence and extent of blowing snow at EGRIP (or comparable locations) is very limited. This statement was based on visual observations reported by co-authors that ran the eddy-covariance measurements in the summers 2016–2019. The statement is updated accordingly (ll. 353-354).**

*L314-316: I think this sentence might need rephrasing. I don't see how a simple linear correction function is appropriate given the complexity and non-linearity you just described.*

**We agree that the formulation in the original version of the article was too brief. In fact, the applied correction function is of linear form (LHF_corrected=m*LHF+b), however it is linearly corrected using the saturation specific humidity, which itself is non-linearly related to the temperature. We updated the text accordingly to provide clarification (ll. 377-380).**

*L316-321: Again these findings should be clearly qualified. These findings are extremely specific to adjusting one term in a model of mass balance, and not adjusting the implicit impact on other terms accordingly.*

**We added clarification that the correction is only applied offline and has no effect on the simulation itself, as well as the importance of addressing and correcting the temperature biases in MAR (ll. 380-386).**

---

## Author Comment (AC2)

*Responses to the reviewer are given in green.*

*General Comments Reviewer 2:*

*The paper presents an analysis of humidity flux for a site in the accumulation of the Greenland Ice Sheet. Data from summertime direct eddy-covariance (EC) observations, AWS-based bulk aerodynamic model estimates and output from a regional climate model (RCM) simulation are presented and discussed. The bulk estimates are found to give good agreement with summertime EC data. The RCM simulations are shown to have seasonally dependent biases. Correction functions are derived, and the corrected humidity fluxes used to comment on the importance of humidity fluxes for the surface mass balance. The topic should be of high interest to readers of TC and the paper presents a useful combination of new observations and modelling results relevant to the topic. A mix of established and new methods are used, most of which are well described and suitable. The figures are very well made, and the text is concise and mostly clear. Several useful results are presented and discussed. Relevant previous research is cited and discussed. However, further discussion and additional results are needed to support the stated conclusions: While the idea of a scale and offset correction appears to be a useful way to scale the RCM simulations, I would argue the results show that MAR does not capture the all the relevant processes well, particularly in the wintertime. This may limit the applicability of MAR results in other areas, and further discussion is needed. More analysis and discussion of the reasons for the biases in MAR humidity fluxes is needed. This should include the interaction of LW radiation, katabatic winds, blowing snow, and sub-surface temperatures on the profile of temperature and humidity in the lower atmosphere. I am concerned about the exclusion of blowing snow from the MAR simulations, particularly as it has been shown elsewhere to greatly affect the surface humidity flux. The choices made here need further justification, and at least some sensitivity analysis within MAR is needed to show the effect of including blowing snow or not. Further considerations of the limitations of the bulk method for estimating accurate wintertime humidity fluxes, especially in blowing snow conditions. With these additional results and discussion, the paper should make a useful contribution to The Cryopshere.*

**We thank the reviewer for the positive and constructive feedback. We agree that a deeper analysis if the systematic errors in the temperature and, thus, humidity in MAR is needed. However, this is not an easy task and not the focus of this study. In this study, we merely aim to estimate the potential impact of these systematic biases on the SMB and the consequent potential underestimation of the role of humidity fluxes in the SMB. It is now stated more clearly that the correction applied in this study has no impact on the simulation itself but is offline. Furthermore, we agree with the reviewer's concern about excluding blowing snow in the simulation and defend and discuss this decision more clearly (see detailed comments below).**

*Specific comments*
*Title – given the above uncertainties in both the bulk and RCM simulations, a more targeted title is warranted e.g. "Correcting regional climate model estimates of humidity flux in the accumulation zone of the Greenland Ice Sheet with in-situ observations."*
**As the aim of this study is not to provide a new, more accurate simulation of the LHF but to point out the importance of the LHF that seems to be underestimated in current MAR simulations, we would prefer to keep the current title.**

*7 – It would be more consistent to present the humidity flux here in mass units (i.e. mm w.eq.)*
**Both values are now given (-1.3 Wm-2, -1.65 mm w.eq.).**

*97 – the explanation given for why blowing snow in MAR was turned off is unclear. Please expand.*
**The blowing snow module was turned off in our analysis for two main reasons: 1.) To keep simplicity and to avoid the potential canceling of two error sources (i.e. temperature biases and systematic errors in the blowing snow simulation), and 2.) to exclude local spacial variations introduced by the blowing snow module from the analysis that can not be expanded to the surroundings of EGRIP due to missing observations. By this approach, the mismatch between the model and the observations can be directly accounted for all error sources, namely, the uncertainty of the observations, the known temperature biases in MAR as well as the neglect of blowing snow in the simulation. Since post-correcting the LHF solely based on the temperature(-gradient) biases in MAR led to a strong improvement of the match between observations and the simulation, this could indicate that blowing snow plays a smaller role at EGRIP than for**

many Antarctic locations. However, this is speculation, and more observations are needed to benchmark the role of blowing snow in EGRIP and to evaluate MAR.

We acknowledge that the discussion of neglecting blowing snow in the simulation has not been comprehensive enough in the original version of this study and we now provide a clearer discussion of the potential impacts on our key results, the correction, and the limits of the applicability of the post-corrected data (ll.393-356).

*127 – please provide further detail of how the setting used to process the EC data. In particular whether any data were excluded or gap filled from the summer records.*

We updated the paragraph with relevant information (ll. XX): We now provide information on the integration time which indeed has been shown to guarantee optimal stationarity and averaged to hourly data to fit the model's frequency. The filter that was applied to the published dataset is a simple noise filter by applying a cut-off threshold of fluxes <-20 W/m2 and >40 W/m2 that is now given in the manuscript but no filtering based on the wind speed or temperatures is applied.

*130 – please either provide additional details and statistics of the EC comparison, or exclude this comment*

Additional information will be provided for the reviewer but since this work is part of a planned publication led by a Master student that is not published yet, we removed the lines in the manuscript.

*162 – earlier you comment that 2019 was warmer and more humid than other years – how does this affect its representativeness.*

Indeed, this is formulated too strongly. By "good representative" we were referring to the fact that the distribution of the LHF in the summer 2019 is very similar to the total distribution in all four summers. Using an example (in Fig. 3 & 6) makes it easier for the reader to understand the different types of systematic errors in winter and summer. We removed the valuation "good" and present the summer 2019 only as an example now (l.XX).

*165 – please clarify the direction of a 'negative' humidity gradient in the text – the definition above only states it is the difference between 2m and surface.*
We updated this (l. 184). A negative humidity gradient refers to a higher specific humidity at the surface compared to 2 m.

*175 – please clarify it is the 'MAR simulated LHF' (as the bulk estimates are a simulation of sorts).*
We updated the line (l. 189).

*198 (also 212 and elsewhere) – it would be clearer to talk about 'offset' and 'scale' biases here – where the summer is primarily an offset bias, whereas in the winter and offset in the magnitude of the flux also produces a bias (difference between means of observations and simulations)*

We like the idea of differentiating the two types of biases and added offset and scale, where applicable.

*214 – how is 0.1 a 'zero-bias'? please revise.*
The offset bias b is normalized in such way that it is exactly zero on January 1st. The average bias for the entire winter (December and January) is 0.1.

*Figure 8 – please provide further detail on how the seasonal curves for the factors m and b are derived?*
b is a function of the monthly averaged surface saturation specific humidity and m a function of its inverse. We post-correct the LHF by applying this simple function in order to estimate both the potential impact of the systematic errors in the MAR simulation and the role that humidity fluxes play in the SMB at EGRIP. We give the exact calculation now in the text (ll. XX) and attached the python code in the Appendix.

*224 – there appears to be an inconsistency in how the annual and summer humidity flux contributions are reported as the text differs (i.e. "In the corrected simulation 5.1% [4% to 6 %] of the annual mass gain (snowfall + deposition) sublimates again," vs "During summer, the amount of sublimated mass corresponds to 31% [26% to 34 %] of the total mass gain". If these are calculated in the same way, consider using the same language for each to avoid confusion.*

**We apologize for the confusion here. In fact, both is calculated the same way and we updated the language to make it sound more similar (ll. 252-253).**

*230 – "Our evaluation shows that MAR captures all relevant processes driving the humidity flux and captures the distribution of the LHF remarkably well (Fig. A8)" I would disagree with this statement. The existence of the biases in the uncorrected simulations highlight that important processes are likely missing from the simulation. But that with correction can be corrected to represent the distribution of humidity flux. Please revise.*

**We agree with the reviewer's critique and removed the statement that all relevant processes are captured but replaced it with a more accurate assessment that both the distribution of the wind speed and the latent heat flux are well captured when an appropriate roughness length is chosen (ll. 262-263).**

*237 – "bias of -1.3Wm−2" it is awkward to introduce an energy term here without context. A mass unit would be better suited.*
**We now give both values (-1.3 Wm-2, -1.65 mm w.eq.), l. 276.**

*242-246 – the distribution of biases in surface and air temperature, and incoming LW needs to be shown here.*
**A figure providing the distribution for surface and 2 m temperature as well as the LW fluxes is added to the Appendix.**

*248 – "systematic error in the LHF is not a bias" as for line 198, the error is a bias, just a different sort of bias – please revise.*
**We completely agree and updated the line (l. 303)**

*255-262 – please show the distribution of biases of variables related to the humidity gradient. These are key elements of evaluation and deserve further description and discussion.*
**See response to the reviewers comment for ll 242-246.**

*291 – how do we know that blowing snow events are rare? The average wind speed is high, and with a dry snow surface, it is likely that blowing snow events are frequent.*

**In fact, information on the occurrence and extent of blowing snow at EGRIP (or comparable locations) is very limited. This statement was based on visual observations reported by co-authors that ran the eddy-covariance measurements in the summers 2016–2019. The statement is updated accordingly (ll. 353-354).**

*Editorial comments*

*249 – "RMSE = 0.73Wm−2" – correct units*
**We corrected the units to m s-1 (l. 304).**

---

## Author Response (AR1)

**Response to the editor's comment**

In this manuscript, we conducted simulations to analyze the latent heat flux. To ensure clarity, we chose to disable the blowing snow module within the MAR model simulation. Both reviewers encouraged us to provide a sensitivity analysis of latent heat flux given this choice. The editor, thus, requested such sensitivity analysis prior to reconsidering our manuscript.

In response to this request, we now present a direct comparison between simulations with and without simulated blowing snow effects for the summer months of 2019 (June and July). Note that this period aligns with the one discussed in more detail within the manuscript. Unfortunately, a change in the computing system prevented us from running the same model version used in the manuscript. Instead, the comparison presented here is based on the latest MAR version (MARv3.14), which may result in slight variations from the exact results in the manuscript.

The figures (Figs 1-3) show that the effects of blowing snow on latent heat flux are minor compared to the differences between the observations and the simulation, as presented in the manuscript, and that the developed correction function is based on.
Specifically, the primary effect is noticeable in the diurnal range of the latent heat flux, while the diurnal mean remains largely unchanged (Fig. 1a, black dash).

Given that it is the diurnal mean latent heat flux that influences the summer surface mass balance, independent of diurnal variations, we conclude that our study's findings remain robust, regardless of the effect of simulating blowing snow. This is true for the EGRIP location (Figs 1-2) as well as for the interior ice sheet where the effects of blowing snow on the latent heat flux commonly stay well below 1 Wm$^{-2}$ (Fig. 3).

It is furthermore worth noting that the blowing snow module within the MAR model has not yet been evaluated against observations on the Greenland Ice Sheet. Given the insensitivity of our results to the application of the module, we have opted to keep the less complex model setup (without blowing snow) simulation in our analysis.

[Figure]

*Fig. 1:  Distribution (a,b) and diurnal cycle (c) of the simulated three-hourly latent heat flux without (green) and with (magenta) the effects of blowing snow simulated in the MAR model for June and July 2019 at the EGRIP location. The black dash in (a) shows the mean value, the thick line shows the median, the box indicates the 25-75$^{th}$ percentile, and the whiskers the 5-95$^{th}$ percentile.*

[Figure]

*Fig. 2: Timeseries of the three-hourly simulated latent heat flux in June and July 2019 at the EGRIP location without (green) and with (magenta) the effects of blowing snow simulated in the MAR model.*

[Figure]

*Fig. 3: Average impact of simulating the effects of blowing snow (BS) in MAR on the latent heat flux (LHF).*